# AI4Water v1.0: An open source python package for modeling hydrological time series using data-driven methods

Ather Abbas[1], Laurie Boithias[2], Yakov Pachepsky[3], Kyunghyun Kim[4], Jong Ahn Chun[5], Kyung Hwa Cho[1]

[1]Urban and Environmental Engineering, Ulsan national institute of science and technology, Ulsan, Republic of Korea.
[2]Geosciences Environnement Toulouse, Université de Toulouse, CNRS, IRD, UPS, 31400 Toulouse, France
[3]Environmental Microbial and Food Safety Laboratory, USDA-ARS, Beltsville, MD, USA.
[4]Watershed and Total Load Management Research Division, National Institute of Environmental Research, Hwangyeong-ro 42, Seogu, Incheon 22689, Republic of Korea
[5]APEC Climate Center, Climate Research Department, Busan, Republic of Korea.

*Correspondence to*: Jong Ahn Chun (jachun@apcc21.org) Kyung Hwa Cho. (khcho@unist.ac.kr)

**Abstract.** Machine learning has shown great promise for simulating hydrological phenomena. However, the development of machine learning-based hydrological models requires advanced skills from diverse fields, such as programming and hydrological modeling. Additionally, data pre-processing and post-processing when training and testing machine learning models is a time-intensive process. In this study, we developed a python-based framework that simplifies the process of building and training machine learning-based hydrological models and automates the process of pre-processing of hydrological data and post-processing of model results. Pre-processing utilities assist in incorporating domain knowledge of hydrology in the machine learning model, such as the distribution of weather data into hydrologic response units (HRUs) based on different HRU discretization definitions. The post-processing utilities help in interpreting the model's results from a hydrological point of view. This framework will help increase the application of machine learning-based modeling approaches in hydrological sciences.

## 1 Introduction

Theory-driven modeling approaches have been traditionally applied to simulate hydrological processes (Remesan and Mathew, 2016). However, with advancements in computation power and data availability, there has been a surge in the application of data-driven approaches to model hydrological processes (Lange and Sippel, 2020). Data-driven approaches that involve time series input data can be used to build several types of hydrological models. Various machine learning approaches have been successfully applied to predict surface water quality (Chen et al., 2020a), estimate stream flow (Shortridge et al., 2016), simulate surface and sub-surface flow (Abbas et al., 2020), forecast evapotranspiration (Ferreira and Da Cunha, 2020), and model groundwater flow and transport (Chakraborty et al., 2020). Deep learning, which includes the application of large neural

networks, has shown promising results for hydrological modeling (Moshe et al., 2020). A typical workflow of data-driven modeling comprises data collection, pre-processing, model selection, training of the algorithm with optimized hyperparameters, and deployment.

Recent advances in the field of data science have resulted in the growth of Python packages, which assist in accomplishing
machine learning and deep learning tasks. According to the latest survey on Kaggle, an online platform for machine learning competitions, the most popular libraries among data scientists are *TensorFlow (Abadi et al., 2016)*, *Pytorch* (Paszke et al., 2019), *Scikit-learn* (Pedregosa et al., 2011), and *XGBoost* (Chen and Guestrin, 2016). These libraries have accelerated research in the field of machine learning owing to their simple user interface and robust implementation of difficult algorithms such as back propagation (Chollet, 2018). However, feature engineering, data pre-processing, and post-processing of results are still
the most time-consuming tasks in building and testing machine learning models (Cheng et al., 2019). Feature engineering includes modifying existing input data and generating new features based on existing data such that it improves learning using data-driven algorithms. This also incorporates background knowledge and context into the model in order to assist the algorithm in learning the underlying function. Infusion of background knowledge, such as basin architecture (Moshe et al., 2020) and land use (Abbas et al., 2020) in data-driven hydrological modeling leverages the algorithm and enhances its
performance (Karpatne et al., 2017). The pre-processing step involves modifying the available data in a form suitable for feeding into the learning algorithm. Nourani et al. (2020) showed how different smoothing and de-noising functions affect the performance of artificial neural networks for forecasting evaporation. The post-processing step includes the calculation of performance metrics, visualization of results, and interpretation.

Recently, several frameworks have been developed to accelerate the process of building and testing machine learning models,
such as *Ludwig* (Molino et al., 2019) and *MLflow* (Zaharia et al., 2018). However, these frameworks are too general and do not deal with the intricacies of time series and hydrological modeling. Several studies have looked at pre-processing, building, training, and post-processing of machine learning models with time series data. These include libraries such as *sktime* (Löning et al., 2019), *Seglearn* (Burns and Whyne, 2018), *Tslearn* (Tavenard et al., 2020), *tsfresh* (Christ et al., 2018), and *pyts* (Faouzi and Janati, 2020). Some libraries have also been developed with a focus on hydrological issues. *Pastas* (Collenteur et al., 2019)
is a library dedicated to analyzing groundwater time series data. *NeuralHydrology* (Kratzert et al., 2019) allows the application of several long short-term memory (LSTM)-based models for rainfall runoff modeling. However, most of these libraries either focus on the processing of data and feature extraction from time series or building and training of the model. A framework that combines pre-processing, feature extraction, building and training, post-processing of model results, and interpretation of data-driven models, particularly for solving hydrological problems, is missing.

For the advancement of machine learning in the field of hydrology, experimentation with readily available and fully documented benchmark datasets is required (Leufen et al., 2021). The collection of hydrological data is usually expensive and time-consuming. Several hydrological datasets are publicly available on different online platforms (Coxon et al., 2020). Although these datasets are documented and organized, they are not usually in a form that can be directly used in machine

learning algorithms. Therefore, there is a need for a uniform and simplified interface to access and feed hydrological data to machine learning algorithms.

In this study, we developed a new framework for fast and rapid experimentation to develop data-driven hydrological models. In this study, we present *AI4Water*, a Python-based framework that assists in machine learning and deep learning-based modeling with a focus on hydrology. The specific objectives of *AI4Water* were to provide a uniform and simplified interface for 1) access and streaming of freely available datasets to data-driven algorithms, 2) pre-processing of hydrological data, 3) automatic feature extraction from hydrological data, 4) automatic model selection and its hyperparameter optimization, and 5) post-processing of results for visualization and interpretation of models.

## 2 Workflow

The core of *AI4Water* is *Model* class, which implements data preparation, building, and training of the model, and makes predictions from the model (Fig. 1). However, *AI4Water* includes several utilities for data pre-processing, feature generation, post-processing and visualization of results, hyperparameter optimization, and model comparison. All of these utilities can be used with *AI4Water* as well as independently. The *Datasets* utility helps in fetching and pre-processing several open-source datasets to be used in machine learning models. The *SpatialProcessing* utility allows distribution of weather data among hydrologic response units (HRUs) using different HRU discretization schemes. The *et* sub-module helps calculate potential evapotranspiration using various theoretical methods. The *SeqMetrics* sub-module calculates several time-series performance metrics for regression and classification problems. *HyperOpt* assists in the implementation of various hyperparameter optimization algorithms. The *Experiment* class can be used to compare different machine learning models. Finally, *AI4Water* has an *Interpret* utility that can be used to interpret the model's results.

The *Model* class of *AI4Water* has two implementations and can have three backends. The two implementations are "model-subclassing" and "functional." The backends are either tensorflow, pytorch, or none of them. The backends, together with the implementations, determine the attributes that the *Model* class will inherit upon its creation. In model-subclassing implementation, the *Model* class inherits either from the tensorflow's *Model* class or the nn.module of pytorch. This implementation allows all the attributes from the corresponding backend to be also available from *AI4Water*'s *Model* class. For example, the "*count_params*" attribute of tensorflow's *Model* class can also be obtained from the *AI4Water*'s *Model* class. In functional implementation, the *Model* class of *AI4Water* does not inherit from the parent modules of tensorflow/pytorch. In this case, the built tensorflow/pytorch model object is exposed to the user as a "*_model*" attribute of the *Model* class. This is similar to tensorflow and pytorch libraries, both of which also have model-subclass and functional implementations. For models other than tensorflow or pytorch, the *Model* class does not have any backend. In these cases, the machine learning

models are built using libraries such as scikit-learn, xgboost, catboost, or lightgbm. The built model object is exposed to the user as "*_model*" attribute of the *Model* class.

The success of machine learning is proportional to testing various hypotheses by training and testing machine learning models and analyzing the results (Zaharia et al., 2018). This can quickly lead to a large number of output files. *AI4Water* handles this by automatically saving all the model-related files starting from model creation to pre-processing until post-processing of each output in the respective folders. A detailed output directory structure is shown in Fig. 2. Upon every model run, a directory is created whose name is the date and time when the model is created. This naming convention allows for a simple and distinct directory structure for every new model. This parent directory is called "model path" and contains several sub-folders and files which are related to model configuration, model training, and post-processing of results (Fig. 2a). The results for each target variable are saved in a separate folder. Additionally, the files related to the model's optimized parameters and interpretations are saved in a separate directory. The saved configuration file along with the weights can later be used to reproduce the model's results. In case of hyperparameter optimization, a directory named "hpo path" is created, which consists of several "model paths". Each of these "model paths" correspond to each iteration of the optimization algorithm (Fig. 2b). In case of *Experiments*, when different models are compared, a separate "hpo path" is created for each of the models being compared. Fig. 2c shows the output file structure for an *Experiment* when different machine learning algorithms are compared. This ordered arrangement of results facilitates the fast comparison and analysis of the results.

## 3 Sub-modules and code-structure

The code architecture of *AI4Water*, that is, its sub-submodules, their available classes, and third-party libraries are illustrated in Fig. 3. *AI4Water* comprises 11 sub-modules, among which 10 are task-based, and one is a general-purpose module named "*utils*." These sub-modules can be divided into two categories. The sub-modules on the left-hand side of Fig. 3 are designed for model building, hyperparameter optimization, and model comparison, whereas those on the right-hand side perform pre-processing and post-processing. Each sub-module exposes one or more classes to the user. For example, the *hyperopt* sub-module presents the *Real*, *Categorical*, *Integer*, and *HyperOpt* classes. The third-party libraries required for each sub-module are annotated inside them. There are five "generic" third-party libraries that are required in all sub-modules (lower part of Fig. 3). The *et* and *utils* sub-modules do not require specific third-party libraries and depend only on generic libraries. The arrows in Fig. 3 indicate interaction between the sub-modules. The origin of the arrow denotes the caller sub-module, whereas their end points denote the sub-module that is being called. The *Model* class interacts with the pre-processing and post-processing modules using its functions, the names of which are shown in green in Fig. 3. For example, the *DataHandler* class in the pre-processing sub-module is responsible for data preparation. The *Model* class interacts with *DataHandler* using *training_*data, *validation_data*, and *test_data* methods. These methods are responsible for fetching training, validation, and test data from the *DataHandler* class, respectively.

The large number of utilities in *AI4Water* increases the number of underlying libraries. The *Model* class is built on top of the *Scikit-learn*, *CatBoost*, *XGBoost*, and *LightGBM* libraries to build classical machine learning models. These models have been used in several hydrological studies (Huang et al., 2019; Ni et al., 2020; Shahhosseini et al., 2021). To build deep learning models using neural networks, *AI4Water* uses popular deep learning platforms, such as *TensorFlow* (Abadi et al., 2016) and *Pytorch*. A complete list of the dependencies for *AI4Water* is presented in Table 1. It is divided into two parts. The first half

shows the minimal requirements for running the basic utilities, which include building and training the model and making predictions from it. The second part of Table 1 comprises an exhaustive list of dependencies. These dependencies are required to utilize all functionalities of *AI4Water*. However, these utilities are optional and do not hamper the basic package functionality. Moreover, the modular structure of *AI4Water* allows the user to install libraries corresponding to a particular sub-module while ignoring the others, which are not required. For example, in order to use the *HyperOpt* class for

hyperparameter optimization, libraries related to post-processing are not required. Table 1 also presents the exact version of the underlying libraries, which were used to test the 1.0 version of *AI4Water*. *AI4Water* handles the version conflicts of the underlying libraries, thereby making it version-independent. This implies that the user can use any version greater than the version number provided in Table 1.

### 3.1 Datasets

The first step in building a data-driven hydrological model is to obtain the data. There have been several efforts by the hydrological science community to build hydrological datasets that are publicly available. For example, for rainfall-runoff modeling, there exists the CAMELS dataset for several countries (Addor et al., 2017). The CAMELS dataset consists of daily weather data and streamflow records for multiple catchments. Another large rainfall runoff dataset is LamaH (Klingler et al., 2021), which consists of observations from 859 catchments in Europe. While the number of such open source datasets is large,

the use of these data sources is slow as each database is available on different platforms and implements a different application programming interface (API). A core function of *AI4Water* is to provide a simple and homogeneous API to feed these datasets directly into machine learning models. Fig. 4 shows the usage of the CAMELS_AUS dataset, where the user needs to define only the name of the dataset and the input and output variables. This simple interface will help exploit the use of these datasets. Furthermore, benchmarking open-source datasets will likely accelerate the progress of machine learning in hydrological

science. A brief summary of the rainfall-runoff datasets available in *AI4water* is given in Table 2.

### 3.2 Exploratory data analysis

A crucial step in data-driven hydrological modeling workflow is the visualization of the data. This step assists in understanding the data, finding outliers, selecting relevant features, and guiding the machine-learning-based modeling process. *AI4Water* provides an *eda* sub-module which can be employed to conduct a comprehensive analysis of input and output data. For

example, the correlation plots illustrate the input variables which are more correlated with each other. Heatmaps show the

amount and position of the missing values. Histogram and box-whisker plots depict the distributions of both the input and output variables. This sub-module can also perform a principal component analysis of the input data and plot the principal components. This helps in understanding the dynamics of the input data and filtering the relevant features.

### 3.3 Preprocessing

**3.3.1 Transformations**

Data transformation includes standardizing and transforming the data onto a different scale. Transforming the data can significantly affect the performance of a data-driven model. The *scikit-learn* library provides several transformation functions such as *minmax*, *standardscaler*, *robust*, and *quantile*. Additionally, several decomposition methods such as empirical mode transformation (EMD), ensemble EMD (EEMD), wavelet transform (Sang, 2013), and fast Fourier transform (Sang et al., 165 2009) were found to improve the performance of hydrological models. *AI4Water* provides a uniform interface for all of these transformation methods under the sub-module *Transformations*. The user can apply any of the available transformations to any of the input features by using a simplified and uniform interface. The predicted features are transformed back after the prediction. Fig. 5 shows a comparison of different transformations using a Taylor plot (Taylor, 2001). These results were generated by modeling in-stream *E. coli* concentrations in a small Laotian catchment (Boithias et al., 2021) using LSTM 170 (Hochreiter and Schmidhuber, 1997). The input data was precipitation, relative humidity, air temperature, wind speed and solar radiation.

### 3.3.2 Imputation

Missing values are often found in real-world datasets. However, missing data cannot be fed to machine-learning algorithms. *AI4Water* provides various solutions for handling missing data that can be used using the *impute* method. These include using 175 either the 1) *pandas* library (Mckinney, 2011), 2) scikit-learn library-based methods, or 3) dedicated algorithms to fill the missing input data. The *pandas* library allows the handling of missing values either by filling the missing values using the *fillna* method or interpolating the missing values using the *interpolate* method. Both these methods can be seamlessly used with the *impute* method in *AI4Water*. Several imputation methods for filling missing values are available in the *scikit-learn* library. These methods include *KNNImputer*, *IterativeImputer,* and *SimpleImputer*. *AI4Water* provides a uniform interface for 180 all imputation methods without hindering their functionality.

Several other libraries have been developed that have dedicated algorithms for imputing missing time series data. These include *fancyimpute* (Rubinsteyn and Feldman, 2016) and *transdim* (Chen et al., 2020b). The *fancyimpute* library provides several state-of-the-art algorithms such as *SoftImpute* (Mazumder et al., 2010), *IterativeSVD* (Troyanskaya et al., 2001), *MatrixFactorization*, *NuclearNormMinimization* (Candès and Recht, 2009), and *Biscaler* (Hastie et al., 2015). The *transdim* 185 library provides algorithms based upon neural networks for filling missing data. *AI4Water* provides a simple interface for using these libraries with their full functionalities, using the *impute* method.

### 3.3.3 Missing labels

In supervised machine-learning problems, the training data consist of examples. Each example consists of one or more input data and a corresponding label, which is the true value for the given example. Similar to the input data, it is common for the labels to have missing data. Although the missing values in target features can be handled similarly to that of input features, which has been explained in Section 2.3, this can lead to unrealistic results, particularly when the number of missing values is large. *AI4Water* allows the user to exclude examples with missing labels during model training. For multi-output prediction, one can encounter situations in which all target variables are not available for a given example. *AI4Water* allows the user to handle such situations by masking the missing observations during loss calculation. However, the user can also opt to exclude these examples, although this can reduce the number of examples in water quality problems where the number of samples is already very small.

### 3.3.4 Resampling

Modeling hydrological processes at high temporal resolutions can result in a large amount of data (Li et al., 2021). Training with this large input data can be computationally expensive. However, temporally coarse input data contain little information. *AI4Water* handles large amount of data by either resampling the data at a lower temporal resolution using the *Resample* class, or by skipping every n-th input data, where 'n' represents the time-step. The latter can be achieved by setting the "*input_steps*" argument to a value >1. The default value of this argument is 1, which results in the use of all input data.

### 3.3.5 Feature generation

The incorporation of scientific knowledge into machine learning models is an emerging paradigm for constraining predictions from machine learning models to reality (Wang et al., 2020). The guiding principle of *AI4Water* is to integrate domain-specific knowledge and hydrological data. *AI4Water* automates the calculation of several features and their inputs to the machine learning algorithm. The input data requirement for the calculation of these features is minimal as they are calculated from the raw data. The calculated features are in the form of a time series, which are then directly given as input to machine learning algorithms. The following sections describe the feature generation process in more detail.

**Land use change and HRU discretization**

In rainfall-runoff modeling, the method of discretization of the HRU plays an important role in many theory-driven models such as the Soil and Water Assessment Tool (SWAT) (Neitsch et al., 2011) and Hydrological Simulation Program FORTRAN (HSPF) (Bicknell et al., 1997). An HRU is a building block of a process-driven hydrological model in which all the processes are simulated. The area and formation of an HRU depend on its definition. For example, in the HSPF model, an HRU is defined as a unique land use in a unique sub-basin. On the other hand, the SWAT model considers slope classes and soil type distributions in an HRU. In catchments, which undergo changes in land use over time, the corresponding HRUs also change

with time. Temporal changes in HRUs are a major challenge in most process-driven models (Kim et al., 2018). However, it has been shown that machine learning models can easily incorporate land-use changes with time and dynamic HRU calculations (Abbas et al., 2020). *AI4Water* contains a sub-module *MakeHRUs*, which helps in distributing the time-series of weather data into HRUs using different HRU definitions. Fig. 6 shows two discretization schemes that combine land use, soil type, and sub-basin. However, the user can also add other spatially varying features, such as slope, in the HRU definition. A complete list of the HRU definitions is provided in Table S1. Fig. 7 illustrates the HRU variation with time in a Laotian catchment (Abbas et al., 2020). The HRUs shown in Fig. 7 are defined as a unique land use with a unique soil type. Thus, every HRU has distinct land use and soil characteristics. As there are four land-use types and three soil types in the catchment, the total number of HRUs was 12. We can observe how the area of certain HRUs, e.g., "Alisol_Fallow", decreases with time at the expense of other HRUs (Fig. 7a). The relative contributions of each HRU for the years 2011, 2012, 2013, and 2014 is illustrated in Fig. 7b–e, respectively. The *MakeHRUs* sub-module requires shapefiles of land use, soil and slope to make the HRU according to a given definition.

### 2.3.6 DataHanlder class

The *DataHandler* class prepares the input data for the machine learning model and acts as an intermediate between the *Model* class and other pre-processing classes, such as *Imputation* and *Transformation* classes. The *DataHandler* can read data from various files as long as the data are in a tabular format in those files. The complete list of allowed file types and their accepted file extensions is provided in Table S5. Internally, the *DataHandler* class stores data as a *pandas DataFrame* object, which is a data model of pandas for tabular data (Mckinney, 2011). *DataHandler* can also save processed data as an HDF5 file, which can be used to inspect processed input data.

### 3.4 Evapotranspiration

The amount of evapotranspiration is an important factor that affects the total water budget in a catchment. The impact of evapotranspiration process representation in rainfall-runoff models has been studied extensively (Guo et al., 2017). Several potential and reference evapotranspiration calculation methods are available in the literature. *AI4Water* contains sub-module '*et*' which can be used to calculate the potential evapotranspiration using various methods. These include complex methods such as Penman–Monteith (Allen et al., 1998), which require many input variables, and simplified methods such as Jensen and Haise (Jensen and Haise, 1963), which only depend on temperature. The *et* can furthermore calculate potential evapotranspiration at various time intervals, from 1 min to 1 yr. The names of the 22 evapotranspiration methods available in *et* and their data requirements are summarized in Table S2. The CAMELS Australia dataset (Fowler et al., 2021) comes with pre-calculated potential evapotranspiration using the Morton (Morton, 1983) method. We compared this method with three different potential evapotranspiration calculation methods using *et*, as depicted in Fig. 8.

## 3.5 Hyperparameter optimization

The hyperparameters of a machine learning algorithm are the parameters that remain fixed during model training and significantly influence its performance (Chollet, 2018). Thus, the choice of hyperparameters plays an important role in evaluating the performance of machine learning algorithms. Some of the most popular approaches for optimizing hyperparameters are random search, grid search, and the Bayesian approach. Random search involves randomly selecting parameters from the given space for a given number of iterations. Grid search, on the other hand, comprehensively explores all possible combinations of hyperparameters in the hyperparameter space. Although grid search can ensure global minima, the number of iterations increases exponentially with an increase in the number of hyperparameters. This renders the grid search practically unfeasible for deep neural network-based models, which are computationally expensive. The two commonly used Bayesian approaches are Gaussian processes (Snoek et al., 2012) and the tree of Parzen estimators, (TPE) (Bergstra et al., 2011).

The libraries used to implement these algorithms are *hyperopt* (Bergstra et al., 2013), *scikit-optimize* (Head et al., 2018), *optuna* (Akiba et al., 2019), and *scikit-learn* (Pedregosa et al., 2011). These libraries implement different algorithms with different strengths. The *scikit-optimize* library allows the application of the Bayesian optimization approach using Gaussian Processes. The *scikit-learn* library can be used for random and grid-search-based approaches. The *hyperopt* module assists in Bayesian optimization using TPEs. The *HyperOpt* sub-module in *AI4Water* provides a uniform interface to interact with all of the aforementioned libraries. The integration of *HyperOpt* with its underlying modules not only complements the underlying optimization algorithms but also adds additional functionality, such as visualization. For example, the importance of hyperparameters is plotted using the functional analysis of variance (fANOVA) method proposed by (Hutter et al., 2014).

We demonstrate the use of the *HyperOpt* sub-module of *AI4Water* for optimizing the hyperparameters of an LSTM-based neural network for rainfall-runoff modeling. The input data consisted of climate data, whereas the target was streamflow. For this example, we used CAMELS data from a catchment in Australia (Fowler et al., 2021). We compared the performance of random search, grid search, and two Bayesian algorithms based on Gaussian Processes and TPEs. The convergence plots of all four algorithms are shown in Fig. 9. The Bayesian approach using Gaussian processes was found to be the most useful for minimizing the objective function. The objective function was the minimum of the validation loss. We also observed that grid search, despite a large number of iterations, did not perform better than the other three methods.

## 3.6 Model comparison with Experiment

*AI4Water* consists of an *Experiment* sub-module, which makes it easier to compare different machine learning models. The basic purpose of the *Experiment* class is to compare different models by optimizing their hyperparameters. This is made possible as the *Experiment* class encompasses the *HyperOpt* class, which in turn encompasses the *Model* class (Fig. 10). Thus, the *Experiment* class can be used for combined algorithm selection and hyperparameter optimization (Thornton et al., 2013). The results from the *Experiment* class are organized within an "exp path" directory (Fig. 2). The *Experiment* class can be sub-

classed to compare any number and type of models. It consists of three sub-classed experiments: *MLRegressionExperiment*,
*MLClassificationExperiment*, and *TransformationExperiment*. The *MLRegressionExperiment* class runs and compares approximately 50 different classical machine learning algorithms for a regression task. The *MLClassificationExperiment* class compares classical machine learning algorithms for a classification problem. The *TransformationExperiment* class can be used to compare the application of different transformation techniques (Sect. 2.3.1) on different input and output features.

We conducted an experiment to compare the performance of classic machine learning algorithms in predicting antibiotic-resistant genes (ARGs) at a recreational beach (Jang et al., 2021). The results of this experiment are shown in Fig. 11, which compares the correlation coefficients for the training and test sets. It can be seen that some algorithms can yield an $R^2$ as high as 0.65. Other algorithms provide training $R^2$ as high as 1.0, which indicates overfitting. In particular, we observed strong overfitting in the case of the decision tree regressor and Gaussian process regressor. It can also be inferred from Fig. 11 that ensemble methods such as AdaBoost (Freund and Schapire, 1997), gradient boosting (Friedman, 2001), bagging (Ho, 1998), extra trees (Geurts et al., 2006), and random forest (Liaw and Wiener, 2002) yield better performance than other methods. We also observed that simple linear models such as Lars, Lasso, and multi-layer perceptron are not able to model the dynamic and complex functions of the ARG distribution at the beach. On the other hand, complex non-linear models such as CATBoost (Prokhorenkova et al., 2017), XGBoost (Chen and Guestrin, 2016), and light gradient boosting machines (Ke et al., 2017) are able to adequately capture dynamic features related to the ARG distribution. We also observed that algorithms with cross-validation performed better than their counterparts without cross-validation.

## 3.7 Post processing

The post-processing submodule of ai4water consists of several utilities which can be used once the machine learning model has been trained. These utilities are discussed in detail below.

### 3.7.1 Visualization

The *"visualize"* sub-module, consisting of a *Visualize* class, is used to examine inside the machine learning model. When the model comprises several layers of neural networks, this class plots the outputs of the intermediate layers, gradients of these outputs, weights and biases of intermediate layers, and gradients of these weights. Thus, this class helps to visualize the working of neural networks. It can also be used to plot the decision tree learned by the tree-based machine learning model. We demonstrate the use of this class by building a four-layer neural network to predict streamflow using the CAMELS dataset (Fowler et al., 2021). The four-layered neural network comprises an input layer, two layers of LSTM, and a Dense layer as output layer (Fig. S1). The Dense layer is a fully connected layer which is used for dimensionality reduction (Chollet, 2018). Figures S2–S5 show the outputs of the first LSTM layer and its gradients along with the weights of the first LSTM layer, and the gradients of those weights.

### 3.7.2 Interpretation and Explainable AI

The interpretation of the results of machine learning models is an area of active research. For classical machine learning algorithms, interpretation tools include the plotting of decision trees or input feature importance. For neural network-based models, explainability is considered an even bigger challenge. *AI4Water* consists of a sub-module called *Interpret*, which can be used to plot interpretable results. The *Interpret* class takes the trained model of *AI4Water* as input and plots numerous results, which help to explain the behavior of the model. The exact type of plots generated by the *Interpret* sub-module depends on the algorithm used by the model. For neural network-based models, which consist of a layered structure, the *Interpret* sub-module plots all the trained weights, the outputs of each layer, the gradients of weights, and the gradients of the activations of neural networks. This also includes plotting attention weights if the model consists of an attention mechanism. *AI4Water* automatically plots the results of the model when a model is used for prediction. These include the scatter and line plots of each target variable.

We demonstrate this by using a dual-stage attention model (Qin et al., 2017) for daily rainfall-runoff modeling in catchment number 401203 in the CAMELS Australia dataset (Fowler et al., 2021). The input data consisted of evapotranspiration, precipitation, minimum and maximum temperatures, vapor pressure, and relative humidity. The dual-stage model showed significant performance during training ($R^2 = 0.93$) and test ($R^2 = 0.87$), as shown in Fig. S6. The dual-stage attention model highlights the importance of the input variables for prediction. The attention weights for each of the input variables are shown in Fig. S7–S9. From these figures, we can infer that the highest attention is given to precipitation followed by evapotranspiration. Furthermore, we also observed that the input of the previous 3–4 days was the most important. This can be attributed to the higher attention weights during the first 3–4 lookback steps in these figures. We also observed periodic changes in attention weights for all input variables, which can be attributed to the seasonal variations of input variables.

Several model-agnostic methods have recently been developed to explain black-box machine learning models, such as local independent model explanations (LIME) (Ribeiro et al., 2016) and Shapely Additive Explanations (SHAP) (Lundberg and Lee, 2017). These methods explain the behavior of complex machine-learning models (such as black-box) using a simplified but interpretable model. However, using these methods in high-stake decision-making has been criticized (Rudin, 2019). The explanations of these methods can be local or global. A local explanation describes the behavior of the model for a single example, whereas a global explanation can describe the model's behavior for all examples. The LIME method is only relevant for local explanations, whereas SHAP also provides explanations for approximating the global importance of a feature. *AI4Water* consists of *LimeExplainer* and *ShapExplainer* classes to explain the behavior of machine learning model using the LIME and SHAP methods respectively.

We built an XGBoost (Chen and Guestrin, 2016) model for the prediction of *E. coli* in a Laotian catchment (Boithias et al., 2021). Fig. S10 shows the output of the *LimeExplainer* class, whereas Fig. S11 shows the output of the *ShapExplainer* class. In Fig. S10, a large horizontal bar for a given feature indicates that this feature strongly affected the model's prediction. A positive value indicate that the given feature caused increase in model's prediction. On the other hand, the negative value

indicate that it caused decrease in model's prediction. Thus, large negative value for solar radiation in example 41 indicate that the solar radiation causes large reduction in model's prediction. Large positive values for water level in examples 42 to 46 indicate that the water level in these cases strongly increased model's prediction. The numerical values of features along y-axis indicate which value of feature was responsible for the aforementioned behaviour. Thus, more precisely, the water level above 147.8 causes very large increase in model's prediction. Therefore, we can verify that the *E. coli* prediction during flood events are more strongly affected by water level.

The SHAP module provides more detailed explanation about local as well as global importance of input features on model's prediction. Fig. S11a and Fig. S11b shows the local explanation summary of model in the form of SHAP value of each input feature for each example (Lundberg et al., 2020). Fig. S11a shows that the examples with large SHAP values of water level and suspended matter resulted in large *E. coli* prediction. The $f(x)$ in Fig. S11a indicate sum of SHAP values of all input features.. The prediction of machine learning model is equal to sum of f(x) and base value. The base value is mean of total predictions from model on training data (Lundberg et al., 2018). In our example the base value was 4661.082 MPN100 mL$^{-1}$. The examples in Fig. S11a are clustered in such a way that examples with similar explanations are grouped together. Fig S11b indicate that the large values of water level and suspended particulate matter results in increase in *E. coli.* On the other hand, large values of solar radiation resulted in negative SHAP values. This shows that large solar radiation causes reduction in *E. coli* prediction. Fig S11c shows the global importance of input features for *E. coli* prediction. This global importance is obtained by calculating mean of SHAP value of a feature for all examples (Lundberg and Lee, 2016). The explanations from Fig. S11 correlate with our background understanding of *E. coli* behavior. Several studies have shown that *E. coli* in surface water is strongly affected by suspended solids, water level and solar radiations (Nakhle et al., 2021; Pandey and Soupir, 2013).

### 3.7.3 Performance metrics

Performance metrics are a vital component of the evaluation framework for machine learning (Botchkarev, 2018). There are two major types of performance metrics related to the evaluation of a model's forecasting ability. These include scale-dependent and scale-independent error metrics. Scale-dependent metrics, such as mean absolute error, provide a good estimate of a single model's performance, but they cannot be used across the models because of their scale dependency (Prestwich et al., 2014). Scale-independent error metrics are more useful when comparing the performance of various models (Hyndman and Koehler, 2006). However, certain scale-independent error metrics cannot be defined when one or more observed values are zero, such as percentage errors or relative errors (Hyndman, 2006). The choice of a performance metric to evaluate the model depends on the problem definition and model objectives (Wheatcroft, 2019). *AI4Water* calculates over 100 regression metrics and numerous classification metrics to help the user analyze the general characteristics of the forecasts. These performance metrics are sub-packaged under *SeqMetrics* in *AI4Water*. These metrics are calculated automatically for all the target variables whenever a model is used for prediction using the *predict* method. The metrics are stored in a json file inside

the path of the model (errors.json in Fig. 2). The names of the performance metrics calculated by *AI4Water* are listed in Table
S3. Additionally, several statistical parameters of the predicted variable were calculated and stored in this json file.

## 4 Loading and saving models in a readable json file

All features of *AI4Water* can be accomplished using a configuration file. The configuration file (config.json) of *AI4Water*
consists of a human-readable json file. All the information regarding pre-processing of data, building and training of the model,
predictions, and post-processing of results is written in this file. This file is generated every time a new model is built. One of
380 the advantages of this configuration file is that any user can build and run the models without having to write the code explicitly.
All examples presented in this study can be run using the corresponding configuration files. Fig. 4 shows three examples of
configuration files. Fig. 4a, shows an LSTM-based model built for rainfall-runoff modeling using the CAMELS (Fowler et al.,
2021) dataset. Fig. 4b and c show the usage of the temporal fusion transformer and XGBoost models for the same task. The
user can define commands to control the input and output features to use or the training duration for the model. All
385 hyperparameters of the model can also be set using this configuration file.

## 5. Advanced usage

*AI4Water* was built using the object-oriented programming (OOP) paradigm. Its core logic was implemented by the *Model*
class. The use of OOP allows a user to customize any steps of model building, training, or testing by sub-classing the *Model*
class. This may include the implementation of a custom training loop or a customized loss function. Similarly, the pre-
390 processing and data preparation steps implemented in the *Model* class can also be overwritten for specific usages. For example,
if users want to implement another transformation on the training data, they can subclass the *Model* class and overwrite the
*"training_data"* method. Similarly, the user can customize the loss function by overwriting the "*loss*" method of *Model* class.
Additionally, *AI4Water* exposes the underlying machine learning libraries such as *TensorFlow* and *scikit-learn* to the user.
Thus, users can directly use these libraries and implement the desired configuration. However, this requires a deeper
understanding of the underlying libraries.

## 6. Test coverage and continuous integration

*AI4Water* version 1.0 was tested with continuous integration tools with GitHub Actions to ensure that it passes all the written
tests and can be installed on computers. The tests were conducted on Windows and Linux-based operating systems. In addition,
we tested the package on Python versions 3.6, 3.7, and 3.8. The package was also tested with *TensorFlow* versions 1.15 and
400 above.

**7. Limitations and scope for expansion**

- The current version of *AI4Water* was designed only for supervised learning problems. However, there has been growing interest in unsupervised machine learning models, such as generative adversarial networks (GANs) and reinforcement learning. GANs have been shown to exhibit high performance for time series-related tasks such as filling missing data (Luo et al., 2018) or generating new high-resolution data (Chen et al., 2019). This aspect of GANs can be useful in water quality modeling, where data collection is costly and missing observations are common. Reinforcement learning can be applied to optimal policy design in hydrological systems, such as scheduling the release of water from a dam (Sit et al., 2020).

- Another limitation of *AI4Water* is its dependence on a large number of third-party libraries. This can be challenging during installation when the interdependencies of libraries conflict with each other. Although we have provided the exact versions of the third-party libraries, which were used to test the current version of *AI4Water*, a conflict in future due to the changes in third-party libraries cannot be guaranteed. As *AI4Water* is an open-source project, we consider that such conflicts can be minimized with community inputs.

- *AI4Water* was designed for the rapid testing and experimentation of deep learning models. However, it should be noted that the current version of the framework is not suitable for the deployment of deep learning models in production.

- As all the options to use *AI4Water* are accommodated in a configuration file, this makes it suitable for developing a graphical user interface (GUI. Adding GUIs will further widen the user-base of *AI4Water* by being accessible to non-programmers.

**8. Conclusion**

Modeling hydrological processes by machine learning requires the development of pipelines that encompasses data retrieval, feature extraction, visualization, building, training, and testing the machine learning model, along with visualization and interpretation of its results. The *AI4Water* software introduced in this work was designed to facilitate the development, reuse, and reproducibility of machine learning models for applications in hydrology. *AI4Water* was designed to integrate the domain-specific aspects of hydrological modeling with the professional level of machine learning and data processing software already developed and used by the Python community. We demonstrated the applicability of *AI4Water* with supervised learning examples related to hydrological modeling. Further development of the package is suggested with new features that may make *AI4Water* more versatile. The platform is expected to be practical for a wide range of users interested in hydrological modeling.

**Code and data availability**

The *AI4Water* source code can be found in a publicly available GitHub repository (https://github.com/AtrCheema/AI4Water) and its version 1.0 is archived at https://zenodo.org/record/4904517. The user manual is built into the source code *Docstring* and compiled into a "read the docs" web page (https://ai4water.readthedocs.io/en/latest/) using the Sphinx (Brandl, 2010) software. The Jupyter notebooks replicating the examples described in the manuscript are available in the "examples" directory.

**Team list**

Ather Abbas

Laurie Boithias

Yakov Pachepsky

Kyunghyun Kim

Jong Ahn Chun

Kyung Hwa Cho

**Author contribution**

Ather Abbas: Conceptualization, code development, writing draft

Laurie Boithias: Review and editing

Yakov Pachepsky: Review and editing

Kyunghyun Kim: Review and editing

Jong Ahn Chun: Review and editing, supervision

Kyung Hwa Cho: Conceptualization, Funding acquisition, supervision, review and edition

**Competing interests**

The authors declare that they have no conflict of interest.

**Acknowledgement**

This study was supported by Basic Science Research Program through the National Research Foundation of Korea (NRF) funded by the Ministry of Education (No. 2017R1D1A1B04033074), and Korea Environment Industry and Technology

Institute (KEITI) through the Aquatic Ecosystem Conservation Research Program funded by Korea Ministry of Environment (MOE) (No. 2020003030003). The authors also thank Campus France (PHC STAR 41510WH) for their financial support.


**Tables**

**Table 1. Complete list of third-party Python libraries, which are used by *AI4Water*. The first half the table enlists those libraries which are required while the second half consists of those libraries which are optional.**

| Library Name | Version | Usage |
|---|---|---|
| **numpy** | 1.19.2 | array processing |
| **pandas** | 1.2.4 | array processing |
| **matplotlib** | 3.4.2 | visualization |
| **h5py** | 2.10 | storage |
| **plotly** | 5.0 | extended visualization |
| **tensorflow** | 1.15, 2.1 | building layers of neural networks |
| **scikit-learn** | 0.24.2 | building classical machine learning models |
| **xgboost** | 1.4.2 | implementing XGBoost based algorithms |
| **catboost** | 0.26 | implementing CatBoost based algorithms |
| **lightgbm** | 3.2.1 | implementing Light Gradient Boost based algorithms |
| **Pyspark** | 3.1.2 | Building classical machine learning models |
| **tpot** | 0.11.7 | Optimizing machine learning pipeline |
| **imageio** | 2.9.0 | spatial processing of shape files |
| **shapely** | 1.7.1 | spatial processing of shape files |
| **pyshp** | 0.45 | spatial processing of shape files |
| **Scikit-optimize** | 0.8.1 | Hyperparameter optimization using Bayesian |
| **Optuna** | 2.8.0 | Hyperparameter optimization |
| **hyperopt** | 0.2.5 | Hyperparameter optimization |
| **shap** | 0.39.0 | Model-agnostic interpretation |
| **lime** | 0.2.0.1 | Model interpretation |
| **seaborn** | 0.11.1 | visualization |

**Table 2. Name and attributes of open source datasets included in *AI4Water*.**

| Dataset Name | Number of catchments | Number of Variables | Number of Observations | Location |
|---|---|---|---|---|

| | | | | |
|---|---|---|---|---|
| **CAMELS_AUS** | 222 | 23 | 21184 | Australia |
| **CAMELS_BR** | 593 | 17 | 14245 | Brazil |
| **CAMELS_CL** | 516 | 12 | 38374 | Chile |
| **CAMELS_GB** | 671 | 10 | 16436 | Britain |
| **CAMELS_US** | 877 | 33 | 12784 | United States of America |
| **LamaH** | 859 | 5 | 12775 | Europe |

**Figures**

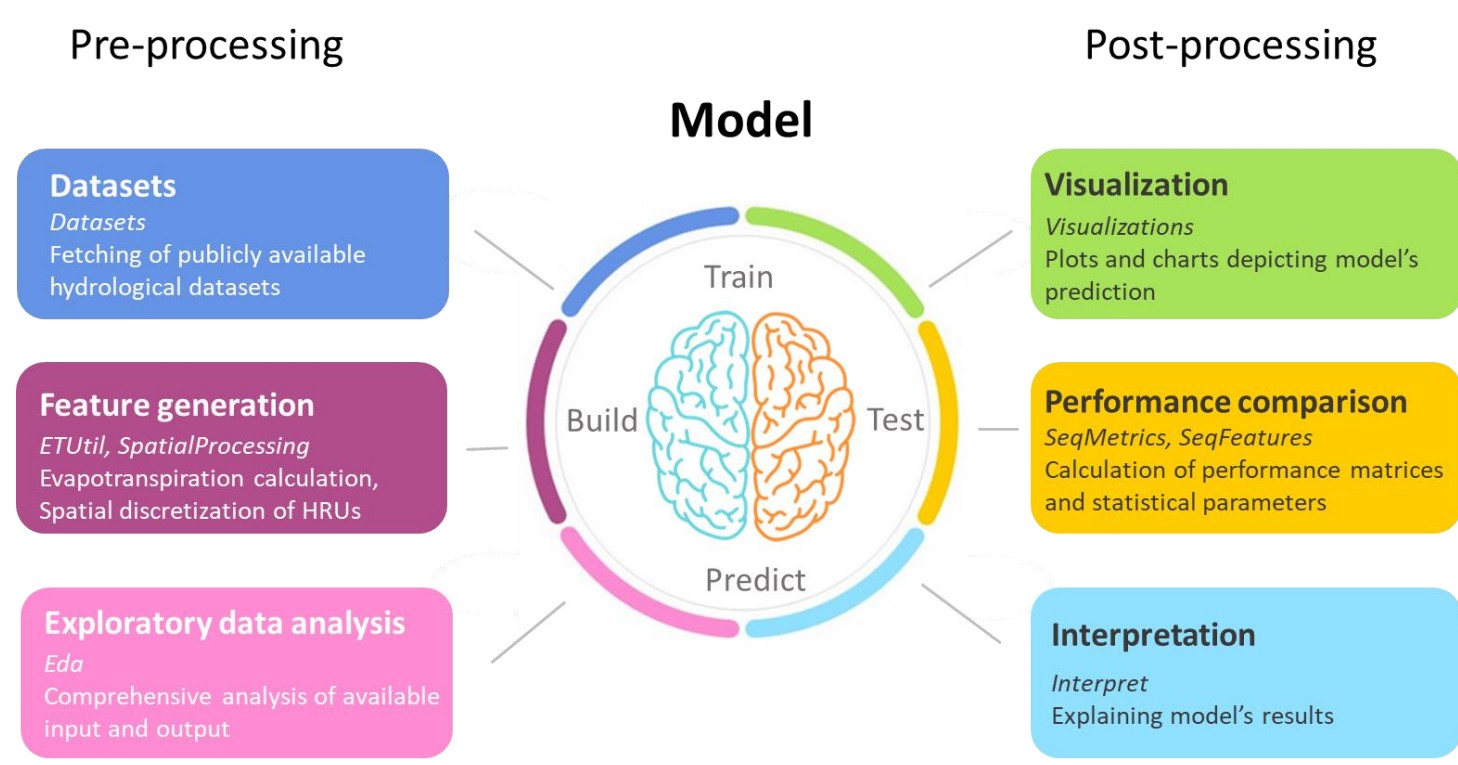

**Figure 1: Conceptual framework of hydrological modeling using *AI4Water*. AI4Water consists of modules for pre-processing and post-processing. The names of the modules are written in italic. The pre-processing steps involve collecting data, conducting exploratory data analysis on data, and generating new features from the data. The core of the model consists of building, training, and predicting. After this step, the predicted steps are used for visualization, performance comparison, and model interpretation.**


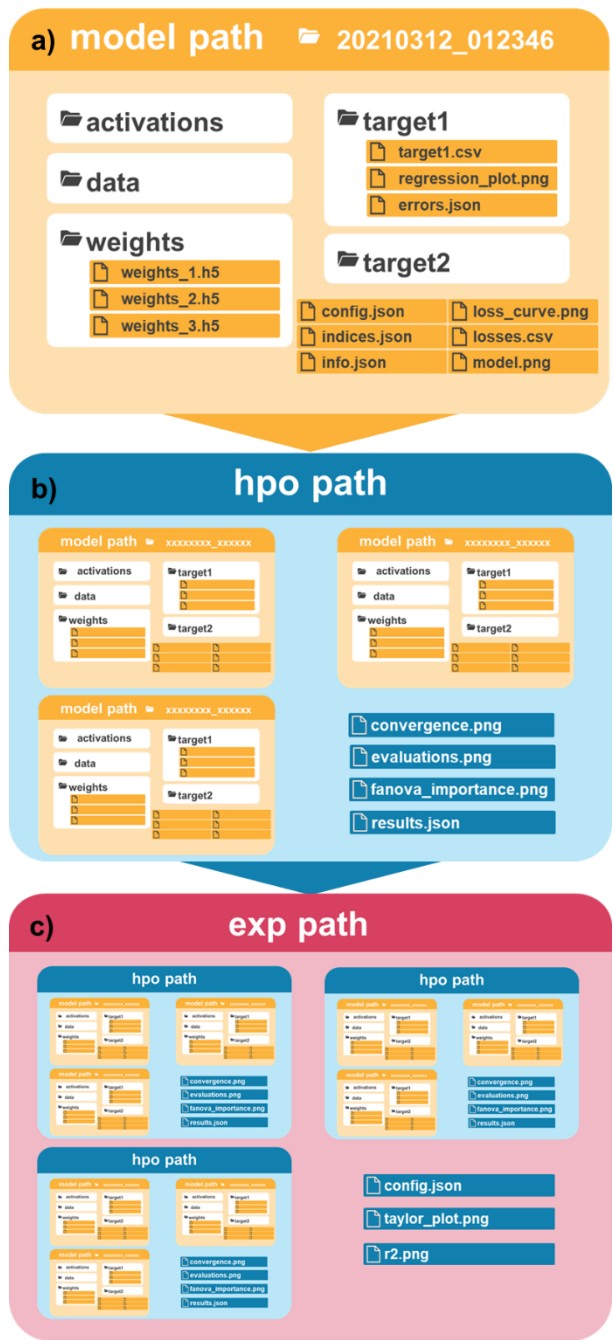

**Figure 2: Output directory structure of *AI4Water*. A "model path" (a) is created upon creation of a new model. An "hpo path" (b) is created during hyperparameter optimization. An "exp path" (c) is created when several models are compared during an experiment. The "hpo path" consists of several "model paths" and an "exp path" consists of several "hpo paths".**

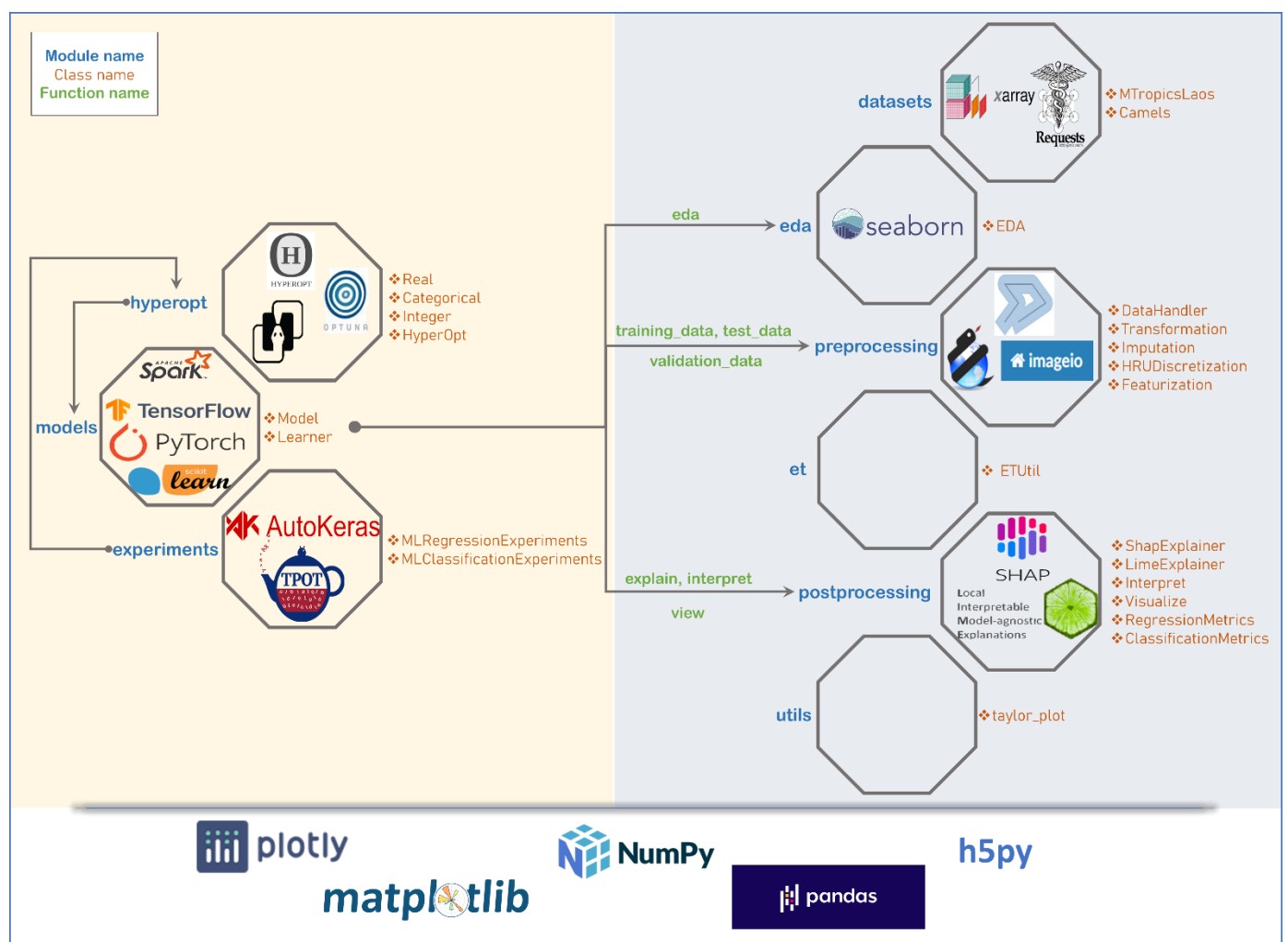

Figure 3: Framework architecture, sub-modules, classes and third-party libraries used by *AI4Water*. Each box represents a sub-module. The names of classes in each sub-module are written along with the corresponding box. The third-party libraries upon which the sub-module depends, are written inside the box. Empty boxes show that these sub-modules do not depend on a specific third-party library. The five generic libraries written at the bottom are used in all sub-modules. Arrows represent the caller sub-module and the sub-module being called. The sub-modules on right hand side are related to pre-processing and post-processing. The *Model* class interacts with pre-processing and post-processing sub-modules using its methods which are written in green colour.

```
{
    "model": {"layers": {"LSTM": 64}},
    "input_features": ['et_mortan', 'precipitation', 'tmax', 'tmin'],
    "output_features": ['streamflow'],
    "data": {"name": "CAMELS_AUS": station": 224206}
}
```

**Figure 4: Examples of declarative model definition in a config.json file. a) shows an example of an LSTM-based model using the CAMELS_AUS data (Fowler et al., 2021). b) and c) show contents of configuration file for using temporal fusion transformer (Lim et al., 2020) and XGBoost (Chen and Guestrin, 2016) for rainfall-runoff modeling using CAMELS_AUS data, respectively.**


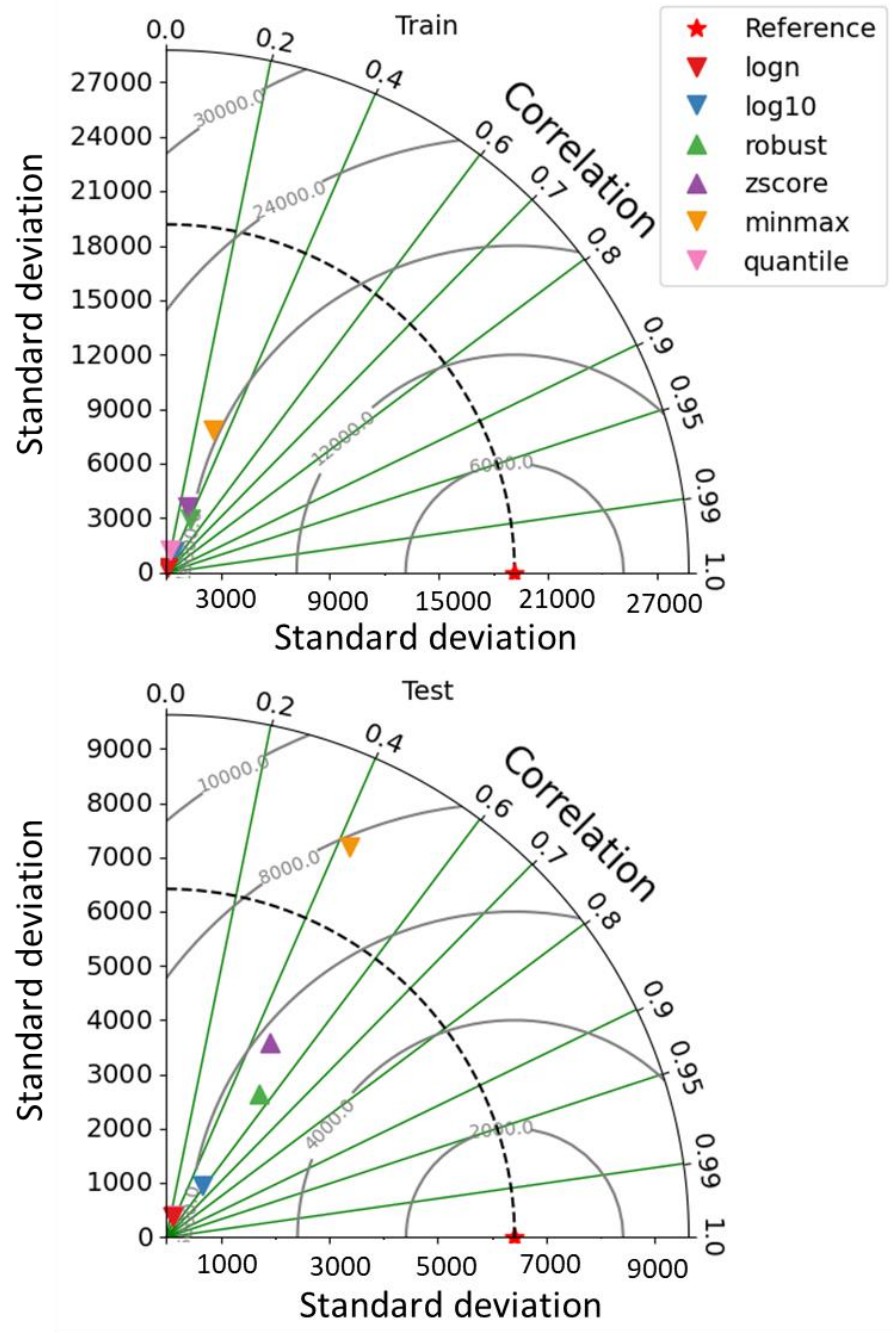

**Figure 5: Comparison of different transformations of output data on the performance of a neural network on the simulation of in-stream *E. coli* concentration (MPN 100 ml) in a watershed in Lao PDR.**

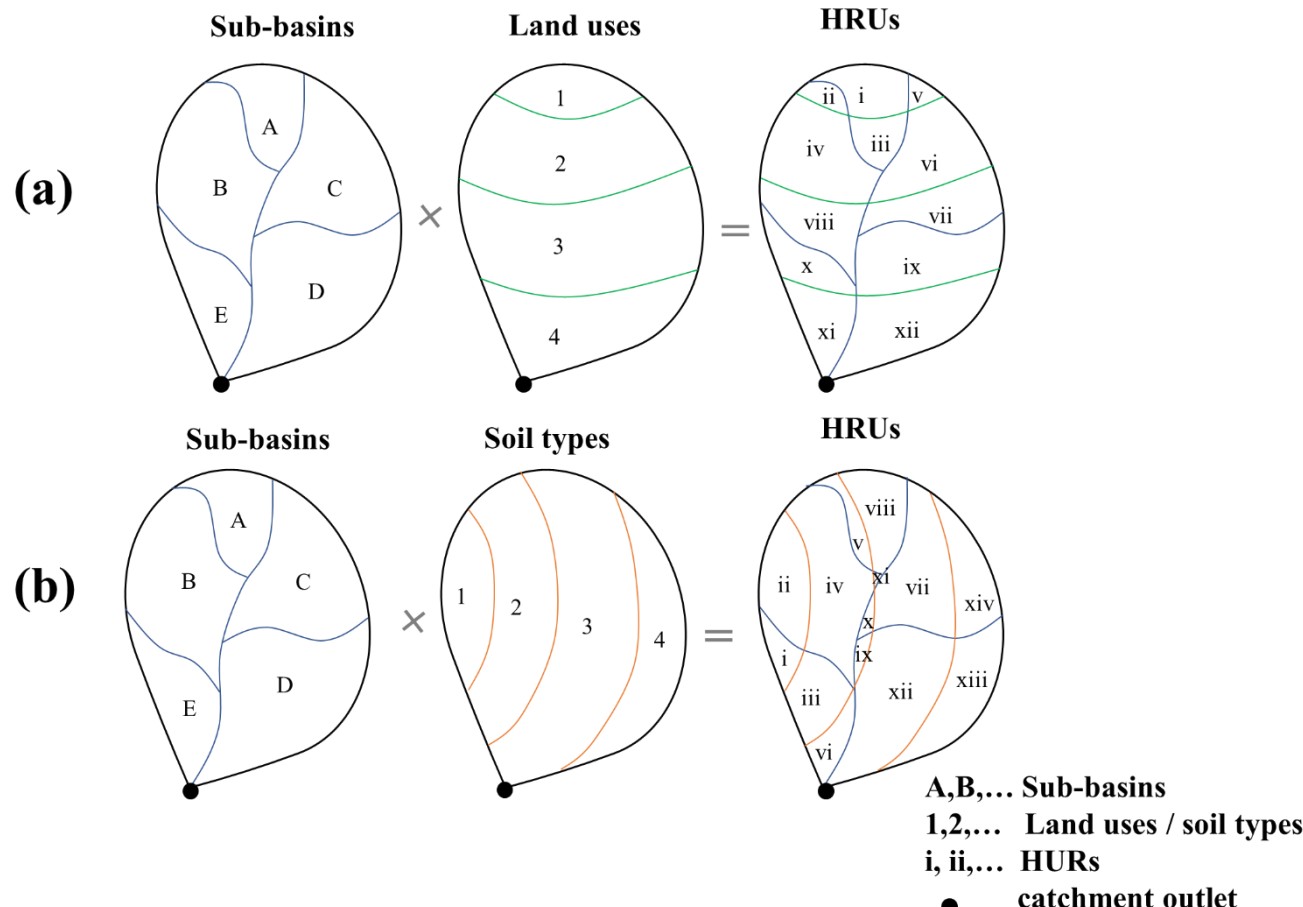

**Figure 6: Example of HRU discretization schemes by combining a): sub-basins and land uses and b) by combining sub-basins and soil types.**

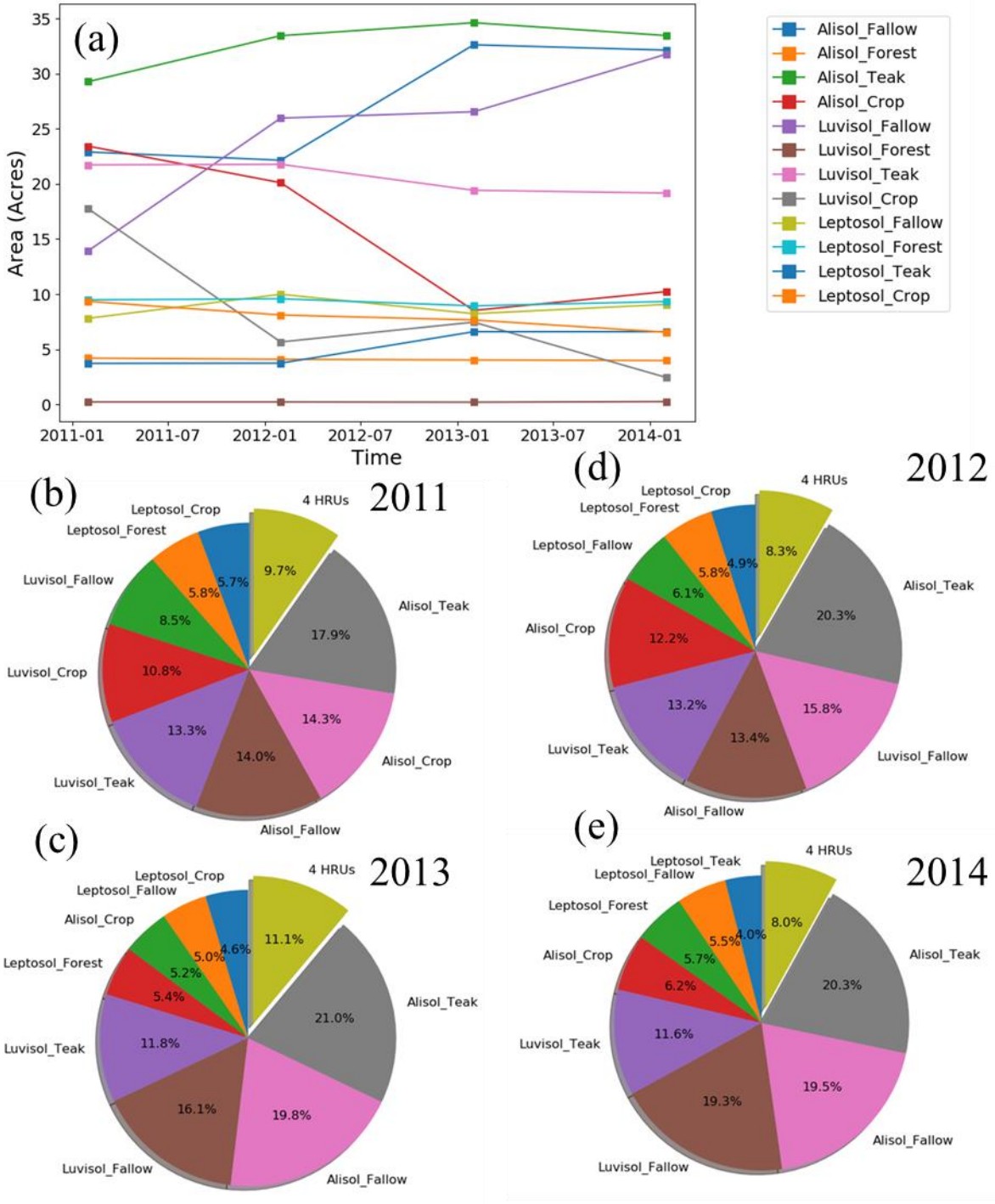

**Figure 7: Discretization of a catchment in Laos (Boithias et al., 2021) according to the HRU definition of "unique land use in unique soil". The catchment consists of three soil types and four land use types. The soil types are Alisol, Luivsol and Leptosol while the**

land use types are Fallow, Forest, Teak, and Crop. The combination of soil types and land use types results in 12 distinct HRUs. (a)
shows annual variation of these 12 land use types while (b)–(e) show the percentage area of HRUs in the catchment in 2011, 2012,
2013, and 2014, respectively.

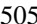

**Figure 8: Comparison of various evapotranspiration methods for the CAMELS_AUS dataset. CAMELS_AUS dataset comes with
Morton method while the remaining three methods are calculated by *et* sub-module of *AI4Water*.**

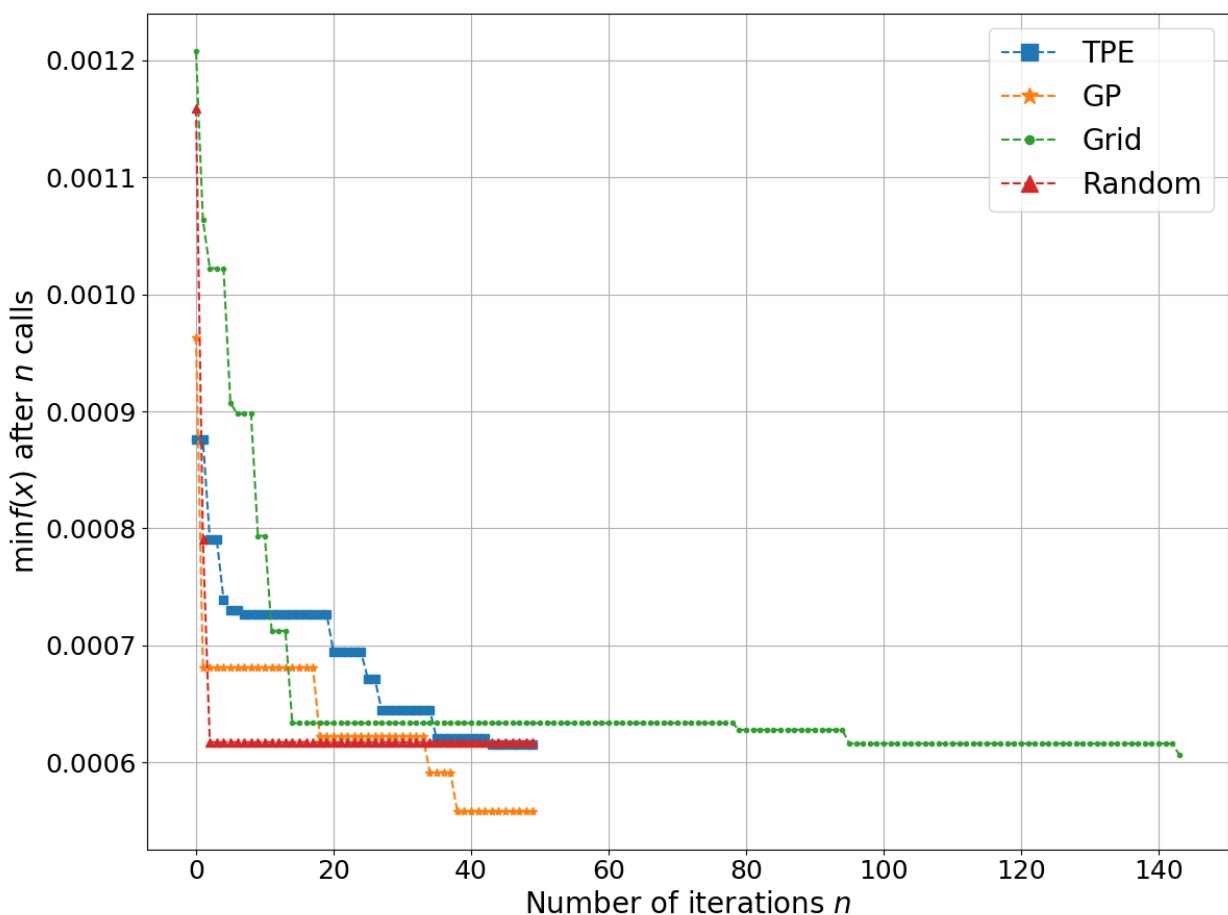

**Figure 9: Comparison of four optimization algorithms for optimizing hyperparameters of an LSTM-based model for rainfall-runoff modeling. GP represents Bayesian with Gaussian Processes while TPE stands for tree of Parzen estimators. Grid and Random stand for grid search and random search-based optimization, respectively. The x-axis shows the number of function evaluations while min f(x) in the y-axis represents the objective function, which takes x hyperparameters and returns the minimum of validation loss.**


# Hierarchy of *AI4Water*

## Experiments

Module for comparison of different models after tuning their hyper-parameters

## HyperOpt

Module for tuning of hyper-parameters of Model using various methods

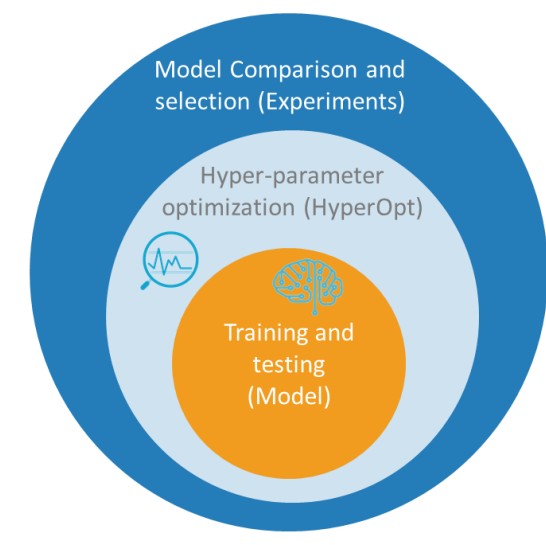

## Model

Module for building, training and prediction of machine learning or deep learning model

**Figure 10: Hierarchy of model building and comparison in *AI4Water*. The Model involves building, training, and prediction. The hyperparameter optimization step iterates over *Model* until the best hyperparameters are obtained. *Experiments* are then designed to compare performance of different model architectures after tuning their hyperparameters.**

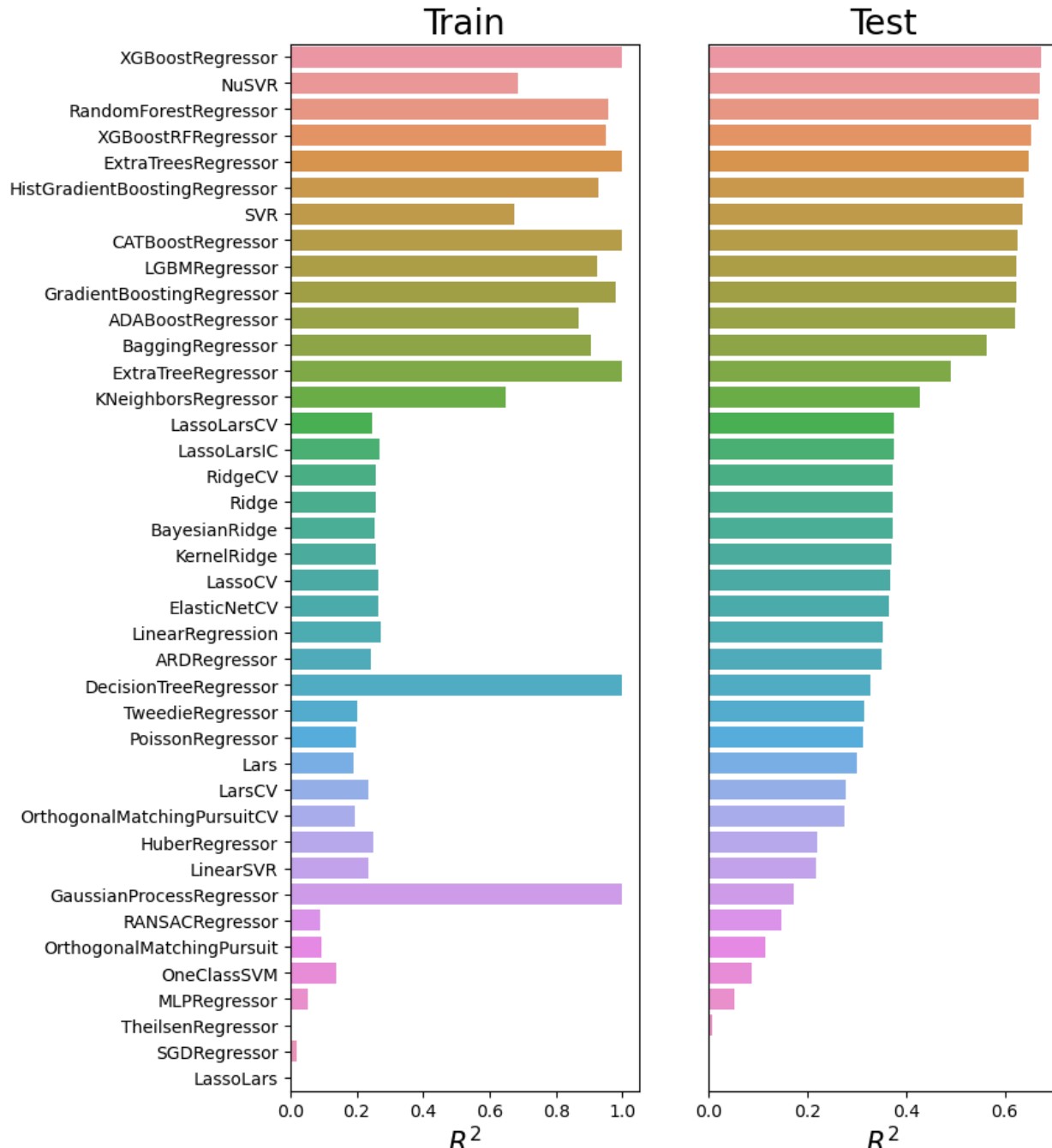

**Figure 11: An 'Experiment' which compares ARG prediction performance at a recreational beach in Korea, using various machine learning algorithms. The y-axis represents abbreviations of the algorithms. The complete names of algorithms are given in Table S4. The hyperparameters of each of the algorithm were optimized during the 'Experiment'.**

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
