# Peer review of "AI4Water v1.0: An open source python package for modeling hydrological time series using data-driven methods"

_Geoscientific Model Development, 2021_

## Author Comment (AC1)

The manuscript describes the new python package AI4Water, intended as a modelling tool for hydrological predictions. It incorporates the basic steps of data-driven analysis and modelling - preprocessing, choosing one or several modelling approaches, post-processing including error analysis and visualization - and makes extensive use of existing python libraries. The focus is strongly on machine learning approaches and the package embraces many of the currently discussed and newly developed algorithms.

The paper is easy to read and, while surely leaving many details out and thus not a manual for the ambitious user, summarizes the fundamental steps in the modelling process in a concise manner. The authors do not develop their own routines or approaches, but the collection of state-of-the-art modelling utilities and approaches is impressive.

**Response:** We thank the reviewer for their valuable comments and suggestions. Your suggestions have substantially improved the quality of the manuscript. Our point-by-point responses to the reviewers' comments are provided below. We have updated the code and archived its latest release at Zenodo (https://zenodo.org/record/5595680). We have updated the documentation of the code that is available at https://ai4water.readthedocs.io/en/latest/. The code to reproduce figures shown in the manuscript is given in the "examples/paper" folder in the code repository.

it would be nice if the authors could address three major issues relevant for anybody intending to analyze their own data:

**Response:** We have revised the manuscript. Accordingly, the code of AI4Water has been assigned to it so that others can analyze their own data using this package. Please find the detailed responses below each comment.

1. It is not obvious how users might get their own data (time series) into the package. The access to two existing databases is implemented, CAMELS and LamaH, and the authors rightly remark that the different data formats, conventions etc. are an obstacle slowing down the analysis process. How generic are the input options to accomodate own data in different formats (text of Excel files, spatially extended time series in netcdf files, and the like)?

**Response:** *AI4Water* contains a *DataHandler* class that pre-processes the input data and prepares the training, validation, and test data. This class can read data from various files as long as the data are in the correct format in those files. *AI4Water* is designed for tabular data; therefore, *DataHandler* expects the data to be in tabular form in a given file. For example, in a csv or excel file, if the data are arranged in a tabular form, one column represents one input or output feature, and each row indicates one example. In this case, the *DataHandler* class reads data from the given file. However, the user must specify the names of the input and output features, which must correspond to the names of columns in the files. Internally, *DataHandler* reads the input file and converts it into a *pandas' DataFrame* object. A *DataFrame* object is a data model of *pandas* for tabular data (Mckinney, 2011). *DataHandler* can read tabular data from the most commonly used file systems, such as a comma separated file (.csv), Microsoft Excel (.xlsx), network common data form (netCDF), feather, parquet (Vohra, 2016), npz, and mat files. If the given file is in netCDF format, it is read using the *xarray* package (Hoyer and Hamman, 2017) and then converted into

pandas *DataFrame*. We have added an ipython notebook named "input_data_file_types.ipynb" to demonstrate this. This notebook shows how users can bring data from .csv, .xlsx, and netcdf files into the *model* using *DataHandler*. The *DataHandler* can save the processed data into an HDF5 file, which can be used by the user for inspection. The processed data comprise training, validation, and test data. In reference to this, we have added the following lines to the manuscript.

**Lines 228 - 233:** The *DataHandler* class prepares the input data for the machine learning model and acts as an intermediate between the *Model* class and other preprocessing classes such as *Imputation* and *Transformation* classes. The *DataHandler* can read data from various files as long as the data are in a tabular format in those files. The complete list of allowed file types and their accepted file extensions is provided in Table S5. Internally, the *DataHandler* class stores data as a *pandas DataFrame* object, which is a data model of pandas for tabular data (Mckinney, 2011). *DataHandler* can also save processed data as an HDF5 file, which can be used to inspect the processed input data.

**Table S5**. File types and their extensions accepted by *AI4Water*.

| File extension | File type |
|---|---|
| **.csv** | Comma separated file |
| **.xlsx** | Microsoft Excel |
| **.npz** | Numpy zipped file |
| **.parquet** | Parquet |
| **.feather** | Feather |
| **.nc** | netCDF5 |
| **.mat** | MATLAB |

2. Expandibility: it might well be that for the specific data at hand or the particular user, other methods then the ones already provided might be desirable. An example would be gap-filling, but also others. The part of the manuscript describing that (chapter 3) is very vague and general, please be more specific.

**Response:** We agree with the reviewer that the ability to customize a certain functionality will be advantageous to the user. Therefore, we used the object-oriented programming (OOP) paradigm for writing the code of *AI4Water*. This paradigm makes it easier to expand or enhance a specific functionality of *AI4Water*, such as customizing the training loop and loss function, or adding an extra pre-processing step to the input data before feeding it to the model. We have also provided examples of codes that customize the loss function, training step, and training loop. These notebooks are available under the examples/paper folder in the code repository. We have added these details in the manuscript in the following lines.

**Lines 381 - 386:** *AI4Water* was built using the object-oriented programming (OOP) paradigm. Its core logic was implemented by the *Model* class. The use of OOP allows a user to customize any steps of model building, training, or testing by sub-classing the *Model* class. This may include the implementation of a custom training loop or a customized loss function. Similarly, the pre-processing and data preparation steps implemented in the *Model* class can also be overwritten for specific usages. For example, if users want to implement another transformation on the training data, they can subclass the *Model* class and overwrite the *"training_data"* method. Similarly, the user can customize the loss function by overwriting the "*loss*" method of *Model* class.

3. Interpretation: it would be wonderful if the package could produce a comprehensive interpretation of the results achieved with the chosen model approaches. Interpretation also

implies making connections to existing hydrologcal knowledge (process understanding) as well as local conditions (metadata) available for the site, its pecularities. However, in this context, interpretation is merely a visualization of the model architecture (e.g. the weights in the case of NNs). A more modest phrasing, e.g. "Model Visualization" instead of "Interpret" as the class name, seems to be more appropriate.

**Response:** We have added a separate sub-module named *"visualize"* to view the model. This sub-module exposes a class named *Visualize* to the user. This class plots the decision tree learned by the tree-based machine learning model. For neural network-based deep learning models, this class can plot the outputs of intermediate layers, weights, gradients of layer outputs, and gradients of weights. This sub-module is separate from the *"Interpret"* and *"Explain"* sub-modules. The *Interpret* sub-module is used to interpret the behavior of attention-based deep learning models, such as DA-LSTM (Qin et al., 2017) or temporal fusion transformers (Lim et al., 2021). The purpose of *"Explain"* sub-module is to explain the output of the machine learning model by considering it as black-box. This sub-module comprises two classes: *ShapExplainer* and *LimeExplainer*, which explain the model using the SHAP (Lundberg and Lee, 2017) and LIME (Ribeiro et al., 2016) methods, respectively. All of these sub-modules are part of the postprocessing sub-module of *AI4water*. We have added figures (S2–S5) using *the Visualize* sub-module of an LSTM model, which show the outputs and weights of LSTM along with the gradients of LSTM outputs and gradients of weights of LSTM. We have added separate sub-sections about interpretability (2.10.1) and visualization (2.10.2) in the manuscript.

**Lines 298 - 305:** The *"visualize"* sub-module, consisting of a *Visualize* class, is used to examine inside the machine learning model. When the model comprises several layers of neural networks,

this class plots the outputs of the intermediate layers, gradients of these outputs, weights and biases of intermediate layers, and gradients of these weights. Thus, this class helps to visualize the working of neural networks and can be used to plot the decision tree learned by the tree-based machine learning model. We demonstrate the use of this class by building a four-layer neural network to predict streamflow using the CAMELS dataset (Fowler et al., 2021). The four-layered neural network comprises an input layer, two layers of LSTM, and one output layer (Fig. S1). Figures S2–S5 show the outputs of the first LSTM layer and its gradients along with the weights of the first LSTM layer, and the gradients of those weights.

[Figure]

**Figure S1:** Architecture of a four-layer neural network used for prediction of streamflow at catchment number 224206 of CAMELS-AUS. Seven climate variables were used and 12 days of historical data was used for training the model. A) The model consisted of 2 LSTM layers followed by a Dense layer as output layer. The output was finally reshaped into 3d array. B) Training and validation loss curves during model training. The model was trained for 700 epochs.

[Figure]

**Figure S2:** Output of first LSTM for 24 days. The model consisted of two LSTM layers with 32 units for each LSTM. The lookback steps indicate the number of historical days used by the model to predict value for next day. The titles for each subplot indicate Julian day for the year 2000.

[Figure]

**Figure S3:** Gradients of outputs of first LSTM for 24 days. The model consisted of two LSTM layers with 32 units for each LSTM. The lookback steps indicate the number of historical days used by the model to predict value for next day. The titles for each subplot indicate Julian day for the year 2000.

[Figure]

**Figure S4:** Weight matrices of LSTM layer. The LSTM layer consists of two weight matrices. The portion of weight matrices responsible for input gate, forget gate, output gate and cell state are highlighted by lack lines.

[Figure]

**Figure S5:** Gradients of weight matrices of LSTM layer. The LSTM layer consists of two weight matrices. The portion of weight matrices responsible for input gate, forget gate, output gate and cell state are highlighted by lack lines.

**Lines 327 – 355:** Several model-agnostic methods have recently been developed to explain black-box machine learning models, such as local independent model explanations (LIME) (Ribeiro et al., 2016) and Shapely Additive Explanations (SHAP) (Lundberg and Lee, 2017). These methods explain the behavior of complex machine-learning models (such as black-box) using a simplified but interpretable model. However, using these methods in high-stake decision-making has been criticized (Rudin, 2019). The explanations of these methods can be local or global. A local explanation describes the behavior of the model for a single example, whereas a global explanation can describe the model's behavior for all examples. The LIME method is only relevant for local explanations, whereas SHAP also provides explanations for approximating the global importance of a feature. *AI4Water* consists of *LimeExplainer* and *ShapExplainer* classes to explain its behavior using the LIME and SHAP methods.

We built an XGBoost (Chen and Guestrin, 2016) model for the prediction of *E. coli* in a Laotian catchment (Boithias et al., 2021). Fig. S10 shows the output of the *LimeExplainer* class, whereas Fig. S11 shows the output of the *ShapExplainer* class. In Fig. S10, a large horizontal bar for a given feature indicate that this feature strongly affected the model's prediction. A positive value indicate that the given feature caused increase in model's prediction. On the other hand, the negative value indicate that it caused decrease in model's prediction. Thus, large negative value for solar radiation in example 41 indicate that the solar radiation causes large reduction in model's prediction. Large positive values for water level in examples 42 to 46 indicate that the water level in these cases strongly increased model's prediction. The numerical values of features along y-axis indicate which value of feature was responsible for the aforementioned behaviour. Thus, more precisely, the water level above 147.8 causes very large increase in model's prediction. Therefore,

we can verify that the *E. coli* prediction during flood events are more strongly affected by water level.

The SHAP module provides more detailed explanation about local as well as global importance of input features on model's prediction. Fig. S11a and Fig. S11b show the local explanation summary of model in the form of SHAP of each input feature for each example (Lundberg et al., 2020). Fig. S11a shows that the examples with large SHAP values of water level and suspended matter resulted in large *E. coli* prediction. The $f(x)$ in Fig. S11a indicate model's prediction. The examples in Fig. S11a are clustered in such a way that examples with similar explanations are grouped together. Fig S11b indicate that the large values of water level and suspended particulate matter results in increase in *E. coli*. On the other, large values of solar radiation resulted in negative SHAP values. This shows that large solar radiation causes reduction in *E. coli* prediction. Fig S11c shows the global importance of input features for *E. coli* prediction. This global importance is obtained by calculating mean of SHAP value of a feature for all examples (Lundberg and Lee, 2016). The explanations from Fig. S11 correlate with our background understanding of *E. coli* behavior. Several studies have shown that *E. coli* in surface water is strongly affected by suspended solids, water level and solar radiations (Nakhle et al., 2021; Pandey and Soupir, 2013).

[Figure]

**Figure S10.** Explanation of XGBoost model for E. coli prediction using LIME method for six selected examples from test data. The explanations show the importance for each input feature by the model.

[Figure]

**Figure S11.** Explanation of XGBoost model for E. coli prediction using SHAP method. The explanations show the importance for each input feature by the model. (a) SHAP values as

heatmap (b) SHAP values for individual examples in test data, (c) global feature importance based upon SHAP values.

The language quality is good to very good with very few typos etc. Some specific comments and corrections:

**Response:** Please find the responses to the specific comments attached below.

l. 78: "time series errors": do you rather mean performance measures rather than errors?

**Response:** Yes, we have replaced the word, "errors" with "performance metrics"

**Lines 78-79:** The *SeqMetrics* sub-module calculates several time-series performance metrics for regression and classification problems.

l. 112: "Fig. 3 shows examples of the three configuration files" -> "Fig. 3 shows three examples for configuration files"

**Response:** We have corrected the sentence.

**Lines 375 - 375:** Fig. 4 shows three examples of configuration files.

l. 118: "obtain large and diverse data" - no, this cannot be guaranteed, and the hope is that modelling is also possible when there is only a limited amount of data from a given catchment, as is often the case!

**Response:** As suggested by the reviewer, we have removed the term "large and diverse" from the sentence.

**Line 138:** The first step in building a data-driven hydrological model is to obtain the data.

l. 139: what is the difference between "scaling" and "transforming the data onto a different scale" ?

**Response:** We agree that the terms "scaling" and "transforming the data onto a different scale'" are similar. Therefore, we have removed the word "scaling" from this sentence.

**Line 159:** Data transformation includes standardizing and transforming the data onto a different scale.

l. 142: EMD is a decomposition, not a transformation method, much like PCA. Of course, using IMFs as input rather than the original variables does change the model setup and has an impact on performance etc. as is correctly stated further down.

**Response:** We have modified the sentence to highlight that EMD is a decomposition method.

**Lines 161-163:** Additionally, several decomposition methods such as empirical mode transformation (EMD), ensemble EMD (EEMD), wavelet transform (Sang, 2013), and fast

Fourier transform (Sang et al., 2009) were found to improve the performance of hydrological models.

l. 146  "were" -> "are"

**Response:** We have replaced the word "were" with "are".

**Line 162:** The predicted features are transformed back after the prediction.

l. 153 "(McKinney, 2011) scikit" -> "(McKinney, 2011), 2) scikit"

**Response:** We have corrected the sentence.

**Lines 172 - 173:** These include using either the 1) *pandas* library (Mckinney, 2011), 2) scikit-learn library-based methods, or 3) dedicated algorithms to fill the missing input data.

ch. 2.4 Missing labels: it should be mentioned that this refers to a classification task only, not to regression.

**Response:** We apologize for the confusion in this section. The absence of target data is common in regression tasks. *AI4Water* can handle these situations for both regression and classification problems.

Also, what is the difference between "exclude examples" (l. 170) and "skip these examples" (l. 173)?

**Response:** The words "exclude" and "skip" mean the same in these lines. Thus, we now use only the term "exclude".

**Lines 192-194:** However, the user can also opt to exclude these examples, although this can reduce the number of examples in water quality problems where the number of samples is already very small.

l. 179 "later" -> "latter"

**Response:** We have replaced "later" with "latter."

**Line 199:** The latter can be achieved by setting the "*input_steps*" argument to a value >1.

l. 198 "time series weather data" -> "time series of weather data"

**Response:** We have replaced "time-series weather data" with "time-series of weather data."

**Line 217-218:** *AI4Water* contains a sub-module *MakeHRUs*, which helps in distributing the time-series of weather data into HRUs using different HRU definitions.

l. 205 how does the user provide HRUs / soil types, land use classes etc. ? Through shapefiles if available?

**Response:** Yes, the module requires shapefiles of soil types, land use classes, slope, and sub-basins to make the HRUs according to a given definition. We have also specified this in the manuscript.

**Line 225 - 226:** The *MakeHRUs* sub-module requires shapefiles of land use, soil and slope to make the HRU according to a given definition.

l. 212 "large" -> "many"

**Response:** We have replaced the word "large" with "many."

**Line 238 - 239:** These include complex methods such as Penman–Monteith (Allen et al., 1998), which require many input variables.

l. 272 "all possible results" -> "many different results"

**Response:** We have corrected the sentence.

**Line 311-312:** The *Interpret* class takes the trained model of *AI4Water* as input and plots numerous results, which help to explain the behavior of the model.

l. 294 "cannot be defined" -  why not?

**Response:** We have mentioned that scale-independent error metrics cannot be defined for some cases. This is true for percentage errors, such as mean absolute percentage error (MAPE), where one or more values in the observed array can be equal to zero. In such cases, the MAPE calculation yields infinity as result. For cases where one or more values are close to zero, the calculated MAPE

values are extremely skewed. This has been emphasized in the literature, such as Hyndman (2006) and Prestwich et al. (2014). We have elaborated this in the manuscript as well.

**Line 362-363:** However, certain scale-independent error metrics cannot be defined when one or more observed values are zero, such as percentage errors or relative errors (Hyndman, 2006).

l. 335 delete the first occurence of "training" in this line

**Response:** We have deleted the first occurrence of 'training.'

**Lines 415 - 417:** Modeling hydrological processes by machine learning requires the development of pipelines that encompasses data retrieval, feature extraction, visualization, building, training, and testing the machine learning model, along with visualization and interpretation of its results

l. 346 Christine, 2014 does not seem to be in the reference list

**Response:** We have corrected this and added a reference for MKDocs in the reference list.

**Lines 425-426:** The user manual is built into the source code *Docstring* and compiled into a "read the docs" web page (https://ai4water.readthedocs.io/en/latest/) using the MKDocs (Christie, 2014) software.

If these comments are taken into account by the authors, the paper should be published by GMD.

**References**

Boithias, L., Auda, Y., Audry, S., Bricquet, J. p., Chanhphengxay, A., Chaplot, V., de Rouw, A., Henry des Tureaux, T., Huon, S., and Janeau, J. l.: The Multiscale TROPIcal CatchmentS critical zone observatory M-TROPICS dataset II: land use, hydrology and sediment production monitoring in Houay Pano, northern Lao PDR, Hydrological Processes, 35, e14126, 2021.

Chen, T. and Guestrin, C.: Xgboost: A scalable tree boosting system, Proceedings of the 22nd acm sigkdd international conference on knowledge discovery and data mining, 785-794, https://doi.org/10.1145/2939672.2939785,

MkDocs. Project documentation with MarkDown.: https://www.mkdocs.org/, last

Fowler, K. J., Acharya, S. C., Addor, N., Chou, C., and Peel, M. C.: CAMELS-AUS: Hydrometeorological time series and landscape attributes for 222 catchments in Australia, Earth System Science Data, 13, 3847-3867, https://doi.org/10.5194/essd-13-3847-2021, 2021.

Hoyer, S. and Hamman, J.: xarray: ND labeled arrays and datasets in Python, Journal of Open Research Software, 5, 2017.

Hyndman, R. J.: Another look at forecast-accuracy metrics for intermittent demand, Foresight: The International Journal of Applied Forecasting, 4, 43-46, 2006.

Lim, B., Arık, S. Ö., Loeff, N., and Pfister, T.: Temporal fusion transformers for interpretable multi-horizon time series forecasting, International Journal of Forecasting, 2021.

Lundberg, S. and Lee, S.-I.: An unexpected unity among methods for interpreting model predictions, arXiv preprint arXiv:1611.07478, 2016.

Lundberg, S. M. and Lee, S.-I.: A unified approach to interpreting model predictions, Proceedings of the 31st international conference on neural information processing systems, 4768-4777,

Lundberg, S. M., Erion, G., Chen, H., DeGrave, A., Prutkin, J. M., Nair, B., Katz, R., Himmelfarb, J., Bansal, N., and Lee, S.-I.: From local explanations to global understanding with explainable AI for trees, Nature machine intelligence, 2, 56-67, https://doi.org/10.1038/s42256-019-0138-9, 2020.

McKinney, W.: pandas: a foundational Python library for data analysis and statistics, Python for high performance and scientific computing, 14, 1-9, 2011.

Nakhle, P., Ribolzi, O., Boithias, L., Rattanavong, S., Auda, Y., Sayavong, S., Zimmermann, R., Soulileuth, B., Pando, A., and Thammahacksa, C.: Effects of hydrological regime and land use on in-stream Escherichia coli concentration in the Mekong basin, Lao PDR, Scientific reports, 11, 1-17, 2021.

Pandey, P. K. and Soupir, M. L.: Assessing the impacts of E. coli laden streambed sediment on E. coli loads over a range of flows and sediment characteristics, JAWRA Journal of the American Water Resources Association, 49, 1261-1269, https://doi.org/10.1038/s41598-017-12853-y, 2013.

Prestwich, S., Rossi, R., Armagan Tarim, S., and Hnich, B.: Mean-based error measures for intermittent demand forecasting, International Journal of Production Research, 52, 6782-6791, 2014.

Qin, Y., Song, D., Chen, H., Cheng, W., Jiang, G., and Cottrell, G.: A dual-stage attention-based recurrent neural network for time series prediction, arXiv preprint arXiv:1704.02971, 2017.

Ribeiro, M. T., Singh, S., and Guestrin, C.: " Why should i trust you?" Explaining the predictions of any classifier, Proceedings of the 22nd ACM SIGKDD international conference on knowledge discovery and data mining, 1135-1144,

Rudin, C.: Stop explaining black box machine learning models for high stakes decisions and use interpretable models instead, Nature Machine Intelligence, 1, 206-215, https://doi.org/10.1038/s42256-019-0048-x, 2019.

Sang, Y.-F.: A review on the applications of wavelet transform in hydrology time series analysis, Atmospheric research, 122, 8-15, 2013.

Sang, Y.-F., Wang, D., Wu, J.-C., Zhu, Q.-P., and Wang, L.: The relation between periods' identification and noises in hydrologic series data, Journal of Hydrology, 368, 165-177, 2009.

Vohra, D.: Apache parquet, in: Practical Hadoop Ecosystem, Springer, 325-335, 2016.

---

## Author Comment (AC2)

The manuscript presents a new framework for fast and rapid experimentation to develop data-driven hydrological models. The manuscript provides an important tool for hydrologic community to utilize machine learning models without expert knowledge. However, there are some minor details missing or not clear in the manuscript.

**Response:** We thank the reviewer for the valuable comments. We have revised the code and manuscript according to the guidelines of the reviewer. Please find the responses to each of the reviewer's comments below. We have updated the code and archived the latest release of the code at Zenodo at the URL https://zenodo.org/record/5595680. We have updated the documentation of the code, which is available at https://ai4water.readthedocs.io/en/latest/. The code to reproduce the figures shown in the manuscript is given in the "examples" folder of the code repository.

Methods section is not easy to follow. It is not clear how the python library is developed, which dependencies are required, underlying methods and protocols for data transfer, processing and visualization.

**Response:** We have added a visual diagram (Fig. 3), which shows the code architecture and the interaction between the different modules of *AI4Water*. We have also changed the numbering and arrangement of the chapters to match the conceptual flowchart shown in Fig. 3. All the sub-modules are now part of Chapter 3. We have also added details about the dependencies, underlying methods of data transfer, and technical details of the framework.

Details about dependencies

**Lines 124 – 136:** The large number of utilities in *AI4Water* increases the number of underlying libraries. The *Model* class is built on top of the *Scikit-learn*, *CatBoost*, *XGBoost*, and *LightGBM* libraries to build classical machine learning models. These models have been used in several hydrological studies (Huang et al., 2019; Ni et al., 2020; Shahhosseini et al., 2021). To build deep learning models using neural networks, *AI4Water* uses popular deep learning platforms, such as *TensorFlow* (Abadi et al., 2016) and *Pytorch*. A complete list of the dependencies for *AI4Water* is presented in Table 1. It is divided into two parts. The first half shows the minimal requirements for running the basic utilities, which include building and training the model and making predictions from it. The second part of Table 1 comprises an exhaustive list of dependencies required to utilize all the functionalities of *AI4Water*. However, these utilities are optional and do not hamper the basic package functionality. Moreover, the modular structure of *AI4Water* allows the user to install libraries corresponding to a particular sub-module while ignoring the others, which are not required. For example, to use the *HyperOpt* class for hyperparameter optimization, libraries related to postprocessing are not required. Table 1 also presents the exact version of the underlying libraries, which were used to test the 1.0 version of *AI4Water*. *AI4Water* handles the version conflicts of the underlying libraries, thereby making it version-independent. This implies that the user can use any version greater than the version number provided in Table 1.

underlying methods and protocols for data transfer

**Lines 228 – 233:** The *DataHandler* class prepares the input data for the machine learning model and acts as an intermediate between the *Model* class and other preprocessing classes, such as *Imputation* and *Transformation* classes. The *DataHandler* can read data from various files as long as the data are in a tabular format in those files. The complete list of allowed file types and their

accepted file extensions is provided in Table S5. Internally, the *DataHandler* class stores data as a *pandas DataFrame* object, which is a data model of pandas for tabular data (Mckinney, 2011). *DataHandler* can also save processed data as an HDF5 file, which can be used to inspect processed input data.

A visual architecture diagram of all classes and third-party libraries will be helpful.

**Response:** We have added a figure depicting all the modules along with their available classes and third-party libraries (Fig. 3). *AI4Water* comprises several sub-modules, such as *eda*, *preprocessing*, *postprocessing*, *datasets*, *et*, Experiments, and *hyperopt*. Two types of third-party libraries are required by *AI4Water*. The first type of libraries are global, which are used in all the modules. These include numpy (Harris et al., 2020), matplotlib (Hunter, 2007), pandas (Mckinney, 2011), h5py (Collette, 2013) and plotly (Sievert, 2020). The second type of libraries are module-specific. Because these modules perform different tasks, their third-party dependencies are different from each other. For example, the "*hyper*_opt" module, which performs hyperparameter optimization, is reliant on hyperopt (Bergstra et al., 2015), scikit-optimize (Head et al., 2018), and optuna (Akiba et al., 2019) libraries. Similarly, the experiment module, which is used to compare different machine learning models, depends on tpot (Olson and Moore, 2016) and auto-keras (Jin et al., 2019) libraries. We have added a section named "sub-modules and code structure," which comprehensively discusses the sub-modules present in *AI4Water* and their interactions with each other.

**Lines 109 – 123:**

**Sub-modules and code-structure**

The code architecture of *AI4Water*, that is, its sub-submodules, their available classes, and third-party libraries are illustrated in Fig. 3. *AI4Water* comprises 11 sub-modules, among which 10 are task-based, and one is a general-purpose module named "*utils*." These sub-modules can be divided into two categories. The sub-modules on the left-hand side of Fig. 3 are designed for model building, hyperparameter optimization, and model comparison, whereas those on the right-hand side perform pre-processing and post-processing. Each sub-module exposes one or more classes to the user. For example, the *hyper_opt* sub-module presents the *Real*, *Categroical*, *Integer*, and *HyperOpt* classes. The third-party libraries required for each sub-module were annotated inside them. There are five "generic" third-party libraries that are required in all sub-modules (lower part of Fig. 3). The *et* and *utils* sub-modules do not require specific third-party libraries and depend only on generic libraries. The arrows in Fig. 3 indicate interaction between the sub-modules. The origin of the arrow denotes the caller sub-module, whereas their end points denote the sub-module that is being called. The *Model* class interacts with the *preprocessing* and *postprocessing* sub-modules using its functions, the names of which are shown in green in Fig. 3. For example, the *DataHandler* class in the *preprocessing* sub-module was responsible for data preparation. The *Model* class interacts with *DataHandler* using *training_*data, *validation_data*, and *test_data* methods, which are responsible for fetching training, validation, and test data from the *DataHandler* class, respectively.

[Figure]

**Figure 3:** Framework architecture, sub-modules, classes and third-party libraries used by *AI4Water*. Each box represents a sub-module. The names of classes in each sub-module are written along with the corresponding box. The third-party libraries upon which the sub-module depends, are written inside the box. Empty boxes show that these sub-modules do not depend on a specific third-party library. The five generic libraries written at the bottom are used in all sub-modules.

Arrows represent the caller sub-module and the sub-module being called. The sub-modules on right hand side are related to pre-processing of data and post-processing of results. The *Model* class interacts with *preprocessing* and *postprocessing* sub-modules using its methods which are written in green color.

Technical details of data integration and API is not provided in detail.

**Response:** We have added technical details about the implementation of the *Model* class and its interaction with other sub-modules of *AI4Water*.

**Lines 83 – 94:** The *Model* class of *AI4Water* has two implementations and can have three backends. The two implementations are "model-subclassing" and "functional." The backends are either tensorflow, pytorch, or none. The backends, together with the implementations, determine the attributes that the *Model* class will inherit upon its creation. In model-subclassing implementation, the *Model* class inherits either from the tensorflow's *Model* class or the nn.module of pytorch. This implementation allows all the attributes from the corresponding backend to be also available from *AI4Water*'s *Model* class. For example, the "*count_params*" attribute of tensorflow's *Model* class can also be obtained from the *AI4Water*'s *Model* class. In functional implementation, the *Model* class of *AI4Water* does not inherit from the parent modules of tensorflow/pytorch. In this case, the built tensorflow/pytorch model object is exposed to the user as a "*_model*" attribute of the *Model* class. This is similar to tensorflow and pytorch libraries, both of which also have model-subclass and functional implementations. For models other than tensorflow or pytorch, the *Model* class does not have any backend. In these cases, the machine learning models are built using libraries such as

scikit-learn, xgboost, catboost, or lightgbm. The built model object is exposed to the user as "*_model*" attribute of the *Model* class.

A discussion about the interaction of *DataHandler* and Model class is also added in the revised manuscript.

**Lines 227 – 233:** The *DataHandler* class prepares the input data for the machine learning model and acts as an intermediate between the *Model* class and other preprocessing classes such as *Imputation* and *Transformation* classes. The *DataHandler* can read data from various files as long as the data are in a tabular format in those files. The complete list of allowed file types and their accepted file extensions is provided in Table S5. Internally, the *DataHandler* class stores data as *a pandas DataFrame* object, which is a data model of pandas for tabular data (Mckinney, 2011). *DataHandler* can also save processed data as an HDF5 file, which can be used to inspect processed input data.

**Table S5**. File types and their extensions accepted by *AI4Water*.

| File extension | File type |
|---|---|
| .csv | Comma separated file |
| .xlsx | Microsoft Excel |
| .npz | Numpy zipped file |
| .parquet | Parquet |
| .feather | Feather |
| .nc | netCDF5 |
| .mat | MATLAB |

Which machine learning framework and version is used in the framework? Does system allow changing or updating the underlying ML library? They are briefly mentioned at the end but they are the most critical components of the framework.

**Response:** The inter-dependence of Python libraries is a complex issue and is difficult to resolve for a novice user. For example, Tensorflow 1.x is compatible with certain versions of a numpy library, whereas Tensorflow 2.x depends on some other version of the numpy. The same is true for other third-party libraries. Therefore, we have specified the exact versions with which the framework was tested. We have also modified the corresponding paragraph to add more details about this.

**Lines 124 – 136:** The large number of utilities in *AI4Water* increases the number of underlying libraries. The *Model* class is built on top of the *Scikit-learn*, *CatBoost*, *XGBoost*, and *LightGBM* libraries to build classical machine learning models. These models have been used in several hydrological studies (Huang et al., 2019; Ni et al., 2020; Shahhosseini et al., 2021). To build deep learning models using neural networks, *AI4Water* uses popular deep learning platforms, such as *TensorFlow* (Abadi et al., 2016) and *Pytorch*. A complete list of the dependencies for *AI4Water* is presented in Table 1. It is divided into two parts. The first half shows the minimal requirements for running the basic utilities, which include building and training the model and making predictions from it. The second part of Table 1 comprises an exhaustive list of dependencies required to utilize all the functionalities of *AI4Water*. However, these utilities are optional and do not hamper the basic package functionality. Moreover, the modular structure of *AI4Water* allows the user to install libraries corresponding to a particular sub-module while ignoring the others, which are not required. For example, to use the *HyperOpt* class for hyperparameter optimization,

libraries related to post-processing are not required. Table 1 also presents the exact version of the underlying libraries, which were used to test the 1.0 version of *AI4Water*. *AI4Water* handles the version conflicts of the underlying libraries, thereby making it version-independent. This implies that the user can use any version greater than the version number provided in Table 1.

**Table 1. Complete list of third-party Python libraries, which are used by *AI4Water*. The first half the table enlists those libraries which are required while the second half consists of those libraries which are optional.**

| Library Name | Version | Usage |
|---|---|---|
| numpy | 1.19.2 | array processing |
| pandas | 1.2.4 | array processing |
| matplotlib | 3.4.2 | visualization |
| h5py | 2.10 | storage |
| plotly | 5.0 | extended visualization |
| tensorflow | 1.15, 2.1 | building layers of neural networks |
| scikit-learn | 0.24.2 | building classical machine learning models |
| xgboost | 1.4.2 | implementing XGBoost based algorithms |
| catboost | 0.26 | implementing CatBoost based algorithms |
| lightgbm | 3.2.1 | implementing Light Gradient Boost based algorithms |
| Pyspark | 3.1.2 | Building classical machine learning models |
| tpot | 0.11.7 | Optimizing machine learning pipeline |
| imageio | 2.9.0 | spatial processing of shape files |
| shapely | 1.7.1 | spatial processing of shape files |
| pyshp | 0.45 | spatial processing of shape files |
| Scikit-optimize | 0.8.1 | Hyperparameter optimization using Bayesian |
| Optuna | 2.8.0 | Hyperparameter optimization |
| hyperopt | 0.2.5 | Hyperparameter optimization |
| shap | 0.39.0 | Model-agnostic interpretation |

| | | |
|---|---|---|
| **lime** | 0.2.0.1 | Model interpretation |
| **seaborn** | 0.11.1 | visualization |

How does the framework keep up with updates in third-party libraries and dependencies used in the framework?

**Response**: In *AI4Water*, we tried to remove the conflicts caused by the changes in the versions of third-party libraries. For example, significant changes were made in tensorflow code from version 1.x to 2.x. However, the user interface for building neural networks in *AI4Water* remained the same for both versions. This is because of the declarative model definition allowed in *AI4Water*. However, *AI4Water* cannot resolve issues that result from the changes in the requirements of third-party libraries. For example, the scikit-optimize library, which implements a Bayesian optimization algorithm, depends on a specific version of the scikit-learn library. This inter-dependency of the third-party libraries is difficult to predict. Similarly, different versions of tensorflow are dependent on different versions of numpy, which can be a major challenge for the user; therefore, we have mentioned the exact versions of all third-party libraries with which this framework has been tested. As this is an open source framework, we expect that future conflicts arising from the dependencies of third-party libraries can also be resolved. We have added the following lines in Chapter 7: Limitations and scope for expansion of the manuscript to highlight this challenge.

**Lines 403 - 408**: Another limitation of *AI4Water* is its dependence on a large number of third-party libraries. This can be challenging during installation when the interdependencies of libraries

conflict each other. Although we have provided the exact versions of the third-party libraries, which were used to test the current version of *AI4Water*, a conflict in future due to the changes in third-party libraries cannot be guaranteed. As *AI4Water* is an open-source project, we consider that such conflicts can be minimized with community inputs.

Does the library allow adding new data transformation, resampling, imputation or other functions?

**Response:** We used the object-oriented programming (OOP) paradigm to build this library. This paradigm allows the customization of any functionality of the *Model* class. The *Model* class interacts with the *DataHandler* class using *training_data*, *validation_data*, and *test_data* methods. Thus, if the users want to implement a custom transformation on the training data, they can subclassify the *Model* class and overwrite the *training_data* function. We have added code examples for implementing custom transformation, customizing the training loop of neural networks, and customizing the loss function. These examples are available as ipython notebooks in the example folder of the code repository. We have also added details regarding this in the manuscript.

**Lines 384 - 386:** For example, if users want to implement another transformation on the training data, they can subclass the *Model* class and overwrite the *"training_data"* method. Similarly, the user can customize the loss function by overwriting the "*loss*" method of *Model* class.

Abadi, M., Barham, P., Chen, J., Chen, Z., Davis, A., Dean, J., Devin, M., Ghemawat, S., Irving, G., and Isard, M.: Tensorflow: A system for large-scale machine learning, 12th {USENIX} symposium on operating systems design and implementation ({OSDI} 16), 265-283,

Akiba, T., Sano, S., Yanase, T., Ohta, T., and Koyama, M.: Optuna: A next-generation hyperparameter optimization framework, Proceedings of the 25th ACM SIGKDD international conference on knowledge discovery & data mining, 2623-2631,

Bergstra, J., Komer, B., Eliasmith, C., Yamins, D., and Cox, D. D.: Hyperopt: a python library for model selection and hyperparameter optimization, Computational Science & Discovery, 8, 014008, 2015.

Collette, A.: Python and HDF5: unlocking scientific data, " O'Reilly Media, Inc."2013.

Harris, C. R., Millman, K. J., van der Walt, S. J., Gommers, R., Virtanen, P., Cournapeau, D., Wieser, E., Taylor, J., Berg, S., and Smith, N. J.: Array programming with NumPy, Nature, 585, 357-362, https://doi.org/10.1038/s41586-020-2649-2, 2020.

Head, T., MechCoder, G. L., and Shcherbatyi, I.: scikit-optimize/scikit-optimize: v0. 5.2, Zenodo, 2018.

Huang, Y., Bárdossy, A., and Zhang, K.: Sensitivity of hydrological models to temporal and spatial resolutions of rainfall data, Hydrology and Earth System Sciences, 23, 2647-2663, 2019.

Hunter, J. D.: Matplotlib: A 2D graphics environment, Computing in science & engineering, 9, 90-95, 2007.

Jin, H., Song, Q., and Hu, X.: Auto-keras: An efficient neural architecture search system, Proceedings of the 25th ACM SIGKDD International Conference on Knowledge Discovery & Data Mining, 1946-1956,

McKinney, W.: pandas: a foundational Python library for data analysis and statistics, Python for high performance and scientific computing, 14, 1-9, 2011.

Ni, L., Wang, D., Wu, J., Wang, Y., Tao, Y., Zhang, J., and Liu, J.: Streamflow forecasting using extreme gradient boosting model coupled with Gaussian mixture model, Journal of Hydrology, 586, 124901, 2020.

Olson, R. S. and Moore, J. H.: TPOT: A tree-based pipeline optimization tool for automating machine learning, Workshop on automatic machine learning, 66-74,

Shahhosseini, M., Hu, G., Huber, I., and Archontoulis, S. V.: Coupling machine learning and crop modeling improves crop yield prediction in the US Corn Belt, Scientific reports, 11, 1-15, 2021.

Sievert, C.: Interactive web-based data visualization with R, plotly, and shiny, CRC Press2020.

---

## Author Response (AR2)

Dear authors,

thank you very much for the revised version of your manuscript. Your revisions addressed almost all of the reviewer comments and your manuscript is close to publication now. Reviewer #1 mentions a few minor issues which I would like you to address in one further revision of your manuscript (see corresponding listing of reviewer #1).

If you have any further questions regarding the revision, do not hesitate to contact me directly.

Thank you and best regards,
Wolfgang Kurtz

**Response:** We thank the editor for these encouraging comments. We have revised the manuscript, supplementary information and figures as per the comments of reviewer 1. Our point-by-point responses to the reviewers' comments are provided below.

Reviewer 1

This revised version presents a significant improvement, including a thorough consideration of the review of the original version. In particular, the framework and code examples are made available to any ambitious reader, such that reproducibility should no longer be an issue, and also the suitability of AI4Water to embrace typical user data is made explicit now.

**Response:** We thank the reviewer for these comments. Please find our point by point response below.

There are only a few minor and mostly trivial points remaining:

- a small number of typos and grammar errors, e.g. l. 115: "categroical" -> "categorical" or line

337: "indicate" -> "indicates" and a few others, in particular in newly added text. Please re-read carefully and correct accordingly.

**Response**: We have corrected these typos in the manuscript. We have also reviewed the manuscript for any further typos and have corrected them.

**Lines 115:** For example, the *hyperopt* sub-module presents the *Real*, *Categorical*, *Integer*, and *HyperOpt* classes.

**Lines 337-338:** In Fig. S10, a large horizontal bar for a given feature indicates that this feature strongly affected the model's prediction.

- Supplement, Fig. S1: could you please explain what a "Dense layer" is? It is nowhere mentioned in the main text and not here as well.

**Response**: A "Dense layer" is a fully connected layer which is a linear multiplication of weights with outputs of preceding layer. The purpose of "Dense" layer is to reduce the dimensions of data to match it to the output size. We have elaborated this in manuscript and updated the following lines.

**Lines 303 – 304:** The four-layered neural network comprises an input layer, two layers of LSTM and a dense layer as output layer (Fig. S1). The Dense layer is a fully connected layer which is used for dimensionality reduction (Chollet, 2018).

- Fig. S3: despite the continous color bar, there seem to be only two colors present - dark red and dark blue - with the exception of day 157. Is there a problem with the range of the colorbar, or some other error here?

**Response:** We thank the reviewer for pointing out this mistake. The binary representation in Fig. S3 is because of selection of very small range of values (vmin and vmax parameters) during plotting. We have corrected this bug in the code. We have also updated Fig. S3. The colors in updated Fig. S3 are continuous and not binary. The updated Fig. S3 is as below

[Figure]

LSTM_0 Gradients of outputs

Lookback steps

LSTM units

- Figure S6: the unit of streamflow can't be "m s^-1", very probably it is "m^3 s^-1", this would also corrspond to "cms" used in Fig. S8 which is not explained but very likely means "m^3 s^-1"

**Response:** We have corrected the units on x and y-axis in Fig. S6 to $m^3s^{-1}$ which stands for cubic meter per second. We also explicitly mentioned in captions of Fig. S6 and Fig. S8.

[Figure]

**Figure S6:** Scatter plots between observed and predicted streamflow for training and test dataset for rainfall-runoff modeling in catchment number '401203' of CAMELS Australia dataset. The units of streamflow are cubic meter per second ($m^3 s^{-1}$).

- Figure S11 (a): there is no scale for f(x), only a horizontal line, presumably indicating zero. Is the model quality related to the distance of f(x) to this zero line? What is f(x)? The explanation "f(x) indicates model's prediction" is very vague.

**Response:** In Fig. S11 (a), the 52 test examples are sorted by the sum of SHAP values over all features. This results in examples with large positive SHAP values on left side and those with large negative SHAP values on right side. The f(x) in Fig. S11 (a) represents sum of SHAP values of all input features for each of the 52 test examples. The horizontal line represents 0 value. The prediction of machine learning model is equal to sum of these SHAP values and base value. This is because, as per Lundberg and Lee (2017), SHAP values explain model's prediction starting from base value. The base value is mean of total predictions from model on training data (Lundberg et al., 2018). In our example the base value was 4661.082 MPN100 mL-1. Negative f(x) for a given example indicates that the sum of SHAP values of all input features for this example is negative. However, the prediction from model is positive because of addition of base value. We have added labels for y-axis to indicate scale of y-axis. We have also modified the caption of this figure to elaborate                                        these                                        points.

[Figure]

**Figure S11.** Explanation of XGBoost model for *E. coli* prediction using SHAP method. The explanations show the

importance for each input feature by the model. (a) SHAP values as heat map. $f(x)$ indicates sum of SHAP values of all input features. The prediction of machine learning model is equal to sum of $f(x)$ and base value. The base value is the sum of models' prediction on training data which was 4661.082. (b) SHAP values for individual examples in test data, (c) global feature importance based upon SHAP values.

The paper is very close to being publishable. The reviewer expresses hope that AI4Water will be used in the hydrology community soon.

**References**

Chollet, F. (2018) Deep learning with Python, Manning Publications Co.

Lundberg, S.M., Erion, G.G. and Lee, S.-I. 2018. Consistent individualized feature attribution for tree ensembles. arXiv preprint arXiv:1802.03888.

Lundberg, S.M. and Lee, S.-I. 2017 A unified approach to interpreting model predictions, pp. 4768-4777.